




# A decade of winter supraglacial lake drainage across Northeast Greenland using C-band SAR

Connor Dean<sup>1</sup>, Randall Scharien<sup>1</sup>, Ian Willis<sup>2</sup>, and Kali McDougall<sup>1</sup>

<sup>1</sup>Department of Geography, University of Victoria, Victoria, BC, Canada

<sup>2</sup>Scott Polar Research Institute, University of Cambridge, Cambridge, UK

\*Correspondence to: Connor Dean (connordean@uvic.ca)

Abstract. This study presents a comprehensive, multi-year assessment of winter supraglacial lake drainages on the Northeast Greenland Ice Sheet, detailing cascading drainage events, examining links to melt-season conditions, and evaluating their potential impact on ice dynamics. Supraglacial lakes can drain rapidly, delivering meltwater to the ice-sheet bed, increasing basal water pressure, reducing friction, and accelerating ice flow. Such drainage events are well-documented across Greenland during the melt season using optical satellite imagery. Recent studies using satellite and airborne radar data reveal that many supraglacial lakes persist beyond summer and may also drain during winter, potentially affecting ice dynamics in a manner similar to melt-season drainages. Here, we use C-band synthetic aperture radar imagery from Sentinel-1 and RADARSAT Constellation Mission spanning ten consecutive winters (2014/2015-2023/2024), to detect winter lake drainages. We develop a normalization method to integrate images from varying acquisition geometries, enabling high-temporal-resolution monitoring. Our analysis identifies 90 winter drainage events from 55 unique lakes, exhibiting substantial interannual variability - from a maximum of 18 events in winter 2018/2019 to a minimum of four events in both 2020/2021 and 2021/2022. Drainages occurred most frequently in early winter, with decreasing frequency as winter progressed. Approximately half of the observed drainages were part of 13 cascading events, each involving two to seven lakes over distances up to ~33 km. Comparisons with preceding melt-season conditions reveal negative correlations between winter drainage frequency and both melt-season intensity and melt-season drainage frequency. Ice velocity analyses over the ten-year period show no sustained seasonal or annual increases attributable to winter drainages, although isolated short-term increases (6-12-day) were observed.

#### 1 Introduction

In recent decades, contributions from the Greenland Ice Sheet (GrIS) to global-mean sea level rise have accelerated due to losses from both surface mass balance (SMB) and glacier dynamics (Mouginot et al., 2019; Shepherd et al., 2020; Box et al., 2022). Supraglacial lakes - hereafter referred to as lakes - are anticipated to expand further inland under continued climate warming (Leeson et al., 2015; Ignéczi et al., 2016, Fan et al., 2025) and contribute to SMB losses by lowering ice sheet surface albedo (Tedesco et al., 2012). In the GrIS ablation zone, lakes form during the melt season and typically drain either slowly, by overtopping their banks and incising outlet channels, or rapidly - within 24 hours - via hydrofracture (Sundal et al., 2009; Selmes et al., 2011; Tedesco et al., 2013). Hydrofracture-driven drainage delivers melt water to the subglacial system via cracks formed by surface stresses, which can enhance basal lubrication and influence ice velocity. These effects are most pronounced in the summer but can persist over annual timescales, particularly in inland regions away from glacier margins (Das et al., 2008; Doyle et al., 2014; Stevens et al., 2015; Christoffersen et al., 2018; Hoffman et al., 2018). At the end of the melt season, lakes may either freeze completely (Selmes et al., 2011; Koenig et al., 2015; Miles et al., 2017) or, as recent studies have shown, drain during winter, when they may trigger ice dynamical responses similar to those observed during the melt season (Schröder et al., 2020; Benedek & Willis, 2021; Maier et al., 2023).






Due to the remoteness and spatiotemporal variability of lakes, Earth observation satellites have become crucial tools for monitoring their evolution and investigating associated processes. Early studies used optical remote sensing for phenological analyses of lake areas and volumes (McMillan et al., 2007; Sneed & Hamilton, 2007; Georgiou et al., 2009; Sundal et al., 2009; Selmes et al., 2011; Johansson et al., 2013; Morriss et al., 2013), a method still widely used due to extensive historical archives. For example, the LANDSAT program dates back to 1972, and the Moderate Resolution Imaging Spectroradiometer (MODIS) sensors, aboard NASA's Terra and Aqua satellites, have provided continuous observations since 2000 and 2002, respectively, enabling multi-year to decadal-scale analyses (Liang et al., 2012; Fitzpatrick et al., 2014; Pope et al., 2016; Williamson et al, 2017; Gledhill & Williamson, 2018; Poinar & Andrews, 2021; Otto et al., 2022).

More recent optical sensors, notably the Sentinel-2A & 2B and WorldView satellites, offer both high spatial resolution (10 m) and a high-temporal frequency (daily), a combination not achievable with earlier platforms (Legleiter et al., 2014; Williamson et al., 2018; Dirscherl et al., 2020; Hochreuther et al., 2021).

While optical imagery has been instrumental in advancing our understanding of lakes during the melt season, the winter period remains understudied since the presence of snow and ice lids, and lack of solar illumination, render optical sensors largely ineffective. Even in summer, optical datasets are prone to temporal aliasing of drainage events, which can bias interpretations of lake drainage mechanisms and timing (Cooley & Christoffersen et al., 2017; Stevens et al., 2024). Addressing these gaps is crucial for developing a more complete, year-round, understanding of lake impacts on ice-sheet dynamics.

Synthetic aperture radar (SAR) sensors transmit electromagnetic radiation in the microwave region of the spectrum, enabling imaging without sunlight and allowing penetration through clouds, as well as through surface snow and ice layers. Incident radar pulses interact with both surface and shallow subsurface (volume) features, producing backscattered signals influenced by surface geometry, roughness, dielectric properties, and SAR system parameters such as frequency, polarization, and incidence angle (Ulaby et al., 1984; Hallikainen et al., 1986; König et al., 2001).

Since the launch of the Copernicus Sentinel-1 mission - with Sentinel-1A in 2014 and Sentinel-1B in 2016 - the use of SAR for lake studies has grown significantly, facilitated in part by the open-access policy of the Copernicus programme. The C-band frequency employed by Sentinel-1, along with past and current missions such as European Space Agency's Envisat-ASAR and the Canadian RADARSAT program, remains the most widely used frequency for observing winter lakes (Johansson & Brown, 2012; Miles et al., 2017; Benedek & Willis, 2021; Zheng et al., 2023).

Initial SAR-focused studies relied on manual delineation and interpretation of winter scenes (Johansson & Brown, 2012; Poinar et al., 2015). Subsequent efforts introduced semi-automated techniques, such as histogram thresholding to delineate lake extents first using a fixed threshold within a single scene (Miles et al., 2017) and later improved via adaptive thresholding in a pan-Greenland application (Zheng et al., 2023). More recent advances include the use of convolutional neural networks (CNNs) trained on dual-polarization (HH-HV) Sentinel-1 imagery (where HH is horizontal transmit/receive and HV horizontal transmit/vertical receive). These CNN-based approaches have enabled regional and ice-sheet-scale mapping of lake distributions during both winter and summer seasons (Dirscherl et al., 2021; Dunmire et al., 2021; Jiang et al., 2022; Zheng et al., 2023).

Several studies have used time series of Sentinel-1 HH and HV backscatter to analyze lake behaviour. Approaches for detecting lake drainage differ from those mapping lake extent, as they rely on identifying anomalies in backscatter between successive images. Schröder et al. (2020) mapped lake extent using a Bayesian classification applied to time-series of Sentinel-1 HH and HH-







HV images and detected winter drainage by tracking backscatter anomalies. Dunmire et al. (2020) combined in-situ ground penetrating radar with Sentinel-1 HH backscatter to identify the winter drainage of a buried lake on the Antarctic ice sheet. In southwest Greenland, Benedek & Willis (2021) used a statistical thresholding method on Sentinel-1 HV backscatter series to detect multi-year winter drainage events. More recently, Hossain et al. (2024) developed a Gaussian Mixture Model-based time series classification to characterize different lake evolution behaviors, including drainage. However, their method classifies lakes by behavior types rather than pinpointing the timing of specific drainage events.

Except for Hossain et al. (2024), who included imagery from multiple orbits and applied temporal smoothing to reduce variability, most studies accounted for the influence of SAR incidence angle on backscatter intensity by using data from a single orbit. This approach ensures consistent backscatter values but limits temporal resolution. With the full Sentinel-1 constellation (Sentinel-1A and Sentinel-1B), the revisit interval is six days; with a single satellite, it increases to twelve. While drainage events are detectable within this range (e.g., Schröder et al., 2020; Benedek & Willis, 2021), accurately resolving the timing of individual events, distinguishing rapid from slow drainage processes, and identifying cascading events, requires finer temporal resolution. Cascading events, where drainage of one lake triggers a chain of subsequent drainages, often occur within days. These chain reactions are driven by the rapid injection of meltwater into subglacial pathways, which generates transient stresses capable of forming crevasses (Christoffersen et al., 2018; Hoffman et al., 2018). Documented across multiple regions of the Greenland Ice Sheet (Liang et al., 2012; Fitzpatrick et al., 2013; Otto et al., 2022; Maier et al., 2023), cascading drainage likely represents a common mechanism of lake drainage.

In this study, we investigate winter drainage events in Northeast Greenland at sub-daily temporal resolution by combining Sentinel 1 images with data from the RADARSAT Constellation Mission (RCM), a three-satellite C-band SAR system launched in 2019. We enhance temporal resolution by applying a novel incidence angle normalization technique that enables the integration of SAR images from multiple overlapping orbits. This method minimizes backscatter variability associated with incidence angle differences - both between orbits and within individual images - allowing for consistent analysis across a wider range of acquisitions.

Using this approach, we identify winter lake drainage events spanning the 2014/2015 to 2023/2024 winter seasons. These include individual lakes that drained as many as six times over the ten-year period, as well as cascading drainage sequences involving up to seven interconnected lakes. We examine interannual variability in the frequency of winter drainage events and assess whether preceding summer melt conditions, such as melt intensity and the number of rapid summer drainages, influence the likelihood of winter drainage. For cascading events, we analyze their sequence, spatiotemporal dynamics, prevalence, and evaluate their proximity to modeled subglacial drainage pathways. Finally, we investigate whether winter drainage events influence ice flow dynamics, assessing both short-term (< 6 to 12 days) velocity changes and longer-term impacts on seasonal and annual ice motion.

# 2 Study area

The study area is located in Northeast Greenland and includes regions below 1500 m elevation on the marine-terminating outlet glaciers Nioghalvfjerdsbræ (79NG) and Zachariæ Isstrøm (ZI), as defined by basin boundaries from Mouginot & Rignot (2019; Fig. 1). The total basin areas of 79NG and ZI are 112,677 km<sup>2</sup> and 92,576 km<sup>2</sup>, of which 16,306 km<sup>2</sup> and 8,704 km<sup>2</sup>, respectively, lie below 1500 m elevation. Near terminus elevations of 79NG and ZI are each approximately 100 m. This area contains one of the most inland-reaching ice streams of the GrIS, transporting ice from the interior to 79NG and ZI at near-terminus velocities of ~1350 m/yr and 1800 m/yr, respectively (Joughin et al., 2016; Fig. 2.1). During the melt season, typically from early June to late

August, extensive networks of supraglacial rivers and lakes form across elevations ranging from ~400 to 1500 m (Sundal et al., 2009; Lu et al., 2021; Hochreuther et al., 2021; Turton et al., 2021; Kanzow et al., 2025).

Northeast Greenland currently accounts for ~18% of the total supraglacial lake volume on the ice sheet, with projections indicating this could increase to 30-35% by the end of the century (Ignéczi et al., 2016). Observations from the Center for Remote Sensing of Ice Sheets (CReSIS) ultra-wideband Snow Radar, flown during NASA's Operation Ice Bridge campaigns, have revealed numerous lakes in the 79NG and ZI basins that persist into late winter - some remaining buried for multiple years (Koenig et al., 2015; Lampkin et al., 2020). Sentinel-1 data have also been used to document widespread lake persistence into the winter months and several suspected drainage events in the region (Schröder et al., 2020). In studies using optical sensors, drainage events have been observed during the summer melt season (Neckel et al., 2020; Hochreuther et al., 2021; Humbert et al., 2025; Lutz et al., 2025). The region is characterized by widespread ice slabs but lacks documented firn aquifers (Miège et al., 2016; Miller et al., 2022; Culberg et al., 2024), which may influence meltwater storage and routing.

Figure 1. Study area in Northeast Greenland. (a) Regional overview map. (b) MEaSUREs multi-year ice sheet velocity mosaic (1995-2015) with sampling points used to extract monthly velocity data from 2015 to 2024 (see Fig. 11; Joughin et al., 2016). (c) Sentinel-1 Extra Wide mode HV-polarized SAR image from October 10, 2016. (d) Landsat RGB mosaic from September 25, 2016. Blue polygons indicate the 10-year melt season lake mask; red outlines show the boundaries of Nioghalvfjerdsbræ (79NG) and Zachariæ Isstrøm (ZI) glaciers (Mouginot & Rignot, 2019).

#### 3 Data and methods

#### 3.1 Optical imagery

Lake locations were identified using Landsat 8 and 9 Operational Land Imager (OLI) imagery acquired annually between late July and the end of August from 2014 to 2023. We used Tier 1 Level 1 Precision Terrain (L1TP) corrected images and filtered for scenes with less than 50% cloud cover. Final image selections were made manually following visual inspection to ensure minimal atmospheric interference. Lake masks were generated using a modified Normalized Water Difference Index modified for ice sheet environments (NDWI<sub>ice</sub>; Yang & Smith., 2013):

$$NDWI_{ice} = \frac{BLUE - RED}{BLUE + RED}$$
 (1)

where BLUE and RED correspond to OLI bands 2 and 4, respectively. NDWI<sub>ice</sub> values between 0.25 and 0.4 have been shown to effectively detect deep water bodies while minimizing the inclusion of slush (Yang & Smith., 2013; Miles et al., 2017; Williamson et al., 2018). We empirically selected a threshold of 0.4, which was found to exclude most slush while reliably identifying deep lakes based on manual inspection of RGB (true-color) images.

False positives caused by clouds and shadows near the ice margin were manually masked out. In addition, supraglacial rivers, characterized by rectilinear or curvilinear shapes, were identified and removed (Lampkin & VanderBerg, 2011; Gledhill & Williamson, 2018). To constrain the study area, we excluded elevations above 1500 m using the ArcticDEM (Porter et al., 2023) and removed land areas using the MEaSUREs Greenland Ice Mapping Project (GIMP) grounded ice mask (Howat et al., 2014; Howat, 2017).

# 145 3.2 Synthetic aperture radar imagery




We used C-band SAR imagery from both Sentinel-1 and RCM, which operate in near-polar, sun-synchronous orbits with a shared center frequency of 5.405 GHz (Table 1). Sentinel-1A was launched in 2014, followed by Sentinel-1B in 2016. When operating together, the two satellites were spaced 180° apart on the same orbit, providing a 6-day revisit interval. Prior to the launch of Sentinel-1B and after its failure in December 2021, Sentinel-1A alone provided a 12-day revisit time. RCM, consisting of three identical satellites spaced 120° apart on a shared orbit, was launched in 2019 as a successor to RADARSAT-2, primarily to serve the operational needs of the Canadian Government (Kroupnik et al., 2021). Following the Sentinel-1B failure, the RCM acquisition strategy was modified to compensate, significantly increasing coverage over the GrIS. In this study, Sentinel-1 imagery is used between 2014 and 2021, and RCM imagery is used thereafter. An overview of sensor specifications is provided in Table 1.

To characterize winter lake drainage activity, we analysed imagery spanning each September to May from 2014/15 through 2023/24. This time frame captures the onset of freeze up, when lakes may be obscured by ice lids and fresh snow, and extends until the onset of surface melt water production. While this period is slightly longer than that used in a comparable study in southwest Greenland (Benedek & Willis, 2021), it is appropriate given the shorter melt season in Northeast Greenland and prior evidence from optical imagery indicating typical lake burial timing (Sundal et al., 2009; Turton et al., 2021; Lutz et al., 2023).

All available Sentinel-1 Ground Range Detected (GRD) images in Interferometric Wide (IW) and Extra Wide (EW) swath modes were acquired from the Alaska Satellite Facility, while RCM GRD imagery in ScanSAR medium-, high-resolution modes and lownoise modes was obtained from the Earth Observation Data Management System (EODMS) operated by Natural Resources Canada




(Table 1). To reduce redundancy, images acquired within two hours of an earlier one (e.g., consecutive ascending or descending passes) were excluded. After this filtering, the mean interval between lake observations across all winters ranged from 2.7 days in 2014/2015 to 0.7 days in 2017/2018.

Image pre-processing was performed using ESA's Sentinel Application Platform (SNAP). The HV polarization band (see Section 3.3 for justification) was calibrated to normalized radar cross-section sigma naught ( $\sigma^{\circ}_{HV}$ ), then terrain-corrected using the ArcticDEM v4.1 (Porter et al., 2023), and reprojected to the NSIDC Sea Ice Polar Stereographic North (EPSG:3413) projection using a 40 m pixel spacing and nearest neighbor resampling. The resulting calibrated and georeferenced data are referred to as  $\overline{\sigma^{\circ}}_{HV}$  hereafter.

**Table 1.** Overview of C-band SAR sensors and modes used. For RCM, ScanSAR50 is the medium-resolution mode, ScanSAR100 is the low-resolution mode, and SCLN is the low-noise mode. Resolution is reported as (r & a), where r = range and a = azimuth.

| Sensor & mode      | Sentinel-1 IW        | Sentinel-1 EW        | RCM ScanSAR50       | RCM ScanSAR100      | RCM SCLN            |
|--------------------|----------------------|----------------------|---------------------|---------------------|---------------------|
| Frequency (GHz)    | 5.405 GHz            | 5.405 GHz            | 5.405 GHz           | 5.405 GHz           | 5.405 GHz           |
| Polarization       | HH + HV              | HH + HV              | HH + HV             | HH + HV             | HH + HV             |
| Resolution (r & a) | 20.0 m x 5.0 m       | 40.0 m x 20.0 m      | 20.0 m x 20.0 m     | 40.0 m x 40.0 m     | 40.0 m x 40.0 m     |
| GRD pixel spacing  | 10.0 m               | 40.0 m               | 20.0 m              | 40.0 m              | 40.0 m              |
| Swath width        | 250 km               | 400 km               | 350 km              | 500 km              | 350 km              |
| Operational years  | April 2014 - present | April 2014 - present | June 2019 - present | June 2019 - present | June 2019 - present |

#### 3.3 Detecting drainage events

C-band SAR, with a typical penetration depth of ~1 to 3 meters in glacial materials (Rignot et al., 2001), is well-suited for detecting water in surface lakes but also shallow subsurface water in buried lakes. In this study, lake drainage events were detected using the HV polarization channel, which is more sensitive to volume scattering and offers deeper penetration than the HH channel (Fischer et al., 2020). This enhanced sensitivity has made it a preferred choice in prior supraglacial lake studies (Miles et al., 2017; Benedek & Willis, 2021; Zheng et al., 2023; Hossain et al., 2024). When a lake drains, the underlying lakebed - composed of firn and glacial ice - is revealed. These materials have a lower dielectric constant than liquid water, and the collapse of an overlying ice lid increases surface roughness. Together, these changes lead to a significant increase in radar backscatter following drainage. In contrast, winter lakes that remain buried beneath snow and ice lids exhibit low backscatter values due to the high dielectric constant of liquid water and the smooth surface interface, which promote specular scattering and suppress radar return (Hallikainen et al., 1986).

# 3.3.1 Melt season supraglacial lake mask

To detect winter drainage events, we merged all summer lake boundaries into a single 10-year composite representing maximum lake extents. Examples of individual lake extents are shown in Fig. 2. For each lake in this mosaic, we tracked  $\overline{\sigma}^{\circ}_{HV}$  throughout the winter seasons. By using a composite rather than separate yearly maps, we accounted for lakes that remain partially or fully buried over multiple years (Koenig et al., 2015; Lampkin et al., 2020), thereby reducing the likelihood of missing winter drainage events due to incomplete seasonal visibility (Benedek & Willis, 2021). Lakes smaller than 0.1 km<sup>2</sup> were excluded from the analysis due to low signal-to-noise ratios, which make backscatter trends less reliable. Additionally, these smaller lakes are more susceptible

to freeze-through during winter, owing to their limited volumes and shallow depth, making winter drainage less likely (Lampkin et al., 2020; Law et al., 2020).

Figure 2. Examples of winter lake behaviour and corresponding Sentinel-1 backscatter time series. Panels show five lake types: (a) a subaerial supraglacial lake that drains; (b) a lake partially covered by a perennial ice lid that drains; (c) a lake with minimal surface expression that drains; (d) a subaerial supraglacial lake that freezes through; and (e) a lake basin that remains dry throughout the melt season and subsequent winter. Column (i) shows Landsat RGB imagery. Columns (ii) and (iii) display C-band SAR images. Red polygons indicate the 10-year Landsat-derived melt season lake masks; yellow dotted polygons show the annual melt season lake masks. Column (iv) shows the time series of mean uncorrected and corrected  $\sigma^0_{HV}$  backscatter from the start (September 1) to the end (May 31) of winter.






#### 3.3.2 Incidence angle correction

The Sentinel-1 and RCM image dataset includes acquisitions from multiple orbital tracks, resulting in incidence-angle-dependent variations in  $\overline{\sigma}^{\circ}_{HV}$  from the same surface target. This angular dependence has been documented in both sea ice and ice sheet studies across various frequencies, polarizations, and also including second-order texture metrics, and is typically addressed by normalizing backscatter or texture to a common incidence angle using linear models (Mahmud et al., 2018; Scharien and Nasonova, 2022; Culberg et al., 2024).

To correct for this effect, we used a least-squares linear regression relationship between  $\overline{\sigma}^{\circ}_{HV}$  and incidence angle. This was done at monthly intervals within the boundaries of 10 non-draining and 10 empty reference lakes per winter season, totaling 200 lakes over the 10-year study period. These reference lakes ranged in size from 95 to 14,678 pixels, with a mean area of 1,290 pixels. Reference lakes were selected based on manual interpretation of winter SAR imagery, optical melt season imagery showing evidence of lake lid collapse (e.g., Schröder et al., 2020; Benedek & Willis, 2021; Maier et al., 2023), and analysis of uncorrected winter backscatter time series.

To account for the influence of surface geophysical changes on regression modelled incidence angle-to-backscatter relationships, before and after drainage events, we applied a multi-slope normalization method. This technique identifies the optimal slope for normalization by evaluating the variance of corrected backscatter values across a sliding window of seven observations. Slope values ranging from 0 to -0.20 dB/° (in -0.005 increments) were tested, and the slope that minimized the variance was selected and applied to correct the center observation in the window to a reference incidence angle of 35°, following:

$$\overline{\sigma}_{\text{HV35}}^{\circ} = \overline{\sigma}_{\text{HV}}^{\circ} + \text{Slope}_{\text{opt}} \cdot (\theta_{\text{ref}} - \theta_{\text{meas}}) \tag{2}$$

Where  $\overline{\sigma^\circ}_{HV35}$  is the backscatter normalized to 35°,  $\overline{\sigma^\circ}_{HV}$  is the uncorrected backscatter, Slope<sub>opt</sub> is the optimal slope  $\overline{\sigma^\circ}_{HV35}$  derived from the variance minimization,  $\theta_{ref}$  is the reference incidence angle of 35°, and  $\theta_{meas}$  is the incidence angle of the observation . Examples of  $\overline{\sigma^\circ}_{HV}$  and  $\overline{\sigma^\circ}_{HV35}$  time series are shown in Fig. 2.

#### 3.3.3 Dynamic threshold and drainage event detection

Despite the incidence angle normalization, residual noise remains in  $\overline{\sigma}^{\circ}_{HV35}$  time series. This noise is primarily attributed to differences in SAR acquisition modes (e.g., Sentinel-1 IW vs. EW), variability across and between ScanSAR sub-swaths, and positional uncertainties. To quantify this residual variability, we used the same set of reference lakes described in Section 3.3.2. For each month across all 10 winters, we calculated all one-, two-, and three-observation cumulative increases in  $\overline{\sigma}^{\circ}_{HV35}$  within the 20 reference lakes. These multi-step increments reflect scenarios in which up to three observations were made over a given lake within a ~36-hour period. The distribution of these short-term increases provides a statistical estimate of the remaining noise in the backscatter time series after normalization.

From this distribution, we derived a dynamic detection threshold defined as four times the monthly median absolute deviation, which we then applied to all lakes within the 10-year composite mask. An increase in  $\overline{\sigma}^{\circ}_{HV35}$  exceeding this threshold was flagged as a potential drainage event. To reduce false positives, an additional filter was applied in which the maximum  $\overline{\sigma}^{\circ}_{HV35}$  value within the 14 days preceding the potential drainage must be lower than the minimum  $\overline{\sigma}^{\circ}_{HV35}$  value with the 14 days following it. When







multiple one-, two-, and three-step cumulative increases in  $\overline{\sigma}^{\circ}_{HV35}$  met these criteria, the start of the drainage window was set to the timestamp of the first observation associated with the largest cumulative increase, while the end of the window was defined as the next local inflection point in the time series.

Each candidate  $\overline{\sigma}^{\circ}_{HV35}$  time series was manually reviewed to remove false positives, most of which were associated with  $\overline{\sigma}^{\circ}_{HV35}$  increases during the freeze-up period (typically September to early October), with inspection of SAR and Landsat imagery used to support the removal. To facilitate analyses of annual and seasonal patterns, we assigned a single representative drainage date to each confirmed event, defined as the midpoint of the automatically-detected drainage window. In cases of suspected cascading drainage, where adjacent lakes appeared to drain in close temporal proximity, we further inspected the  $\overline{\sigma}^{\circ}_{HV35}$  time series to refine the drainage timing. For these events, we selected the first observation at  $\overline{\sigma}^{\circ}_{HV35}$  exceeded the dynamic threshold. The window of most likely drainage was defined by this observation and the one immediately preceding it, providing a more precise estimate of the drainage onset.

# 3.4 Lake depth and volume estimates

We used ArcticDEM strips (Porter et al., 2022) to estimate the depth and volume of lakes prior to winter drainage. These DEM strips are derived from high-resolution optical image pairs using the Surface Extraction with Triangulated Irregular Network-based Search-space Minimization (SETSM), with a vertical uncertainty of 2.0 m, reduced to ~0.2 m after co-registration (Noh & Howat, 2015).

When ArcticDEM strips were available both before and after a drainage within the same winter, we co-registered and differenced them, resulting in a vertical error of 0.28 m. However, such paired DEMs were rare due to sparse temporal coverage - most strips are acquired in late winter or early spring, after the majority of lake drainages occur. For lakes with only post-drainage DEMs, we manually digitized shorelines using the latest available cloud-free Landsat imagery prior to polar night, where visible. We then extracted the mean elevation of the shoreline from the earliest post-drainage DEM as a proxy for the lake lid surface. Subtracting this from the DEM surface yielded lake depth, with an assumed vertical error of 0.2 m. Similar approaches have been used previously for estimating lake volume from high resolution DEMs (Pope et al., 2016; Yang et al., 2019; Maier et al., 2023). If no post-drainage DEM was available for a given winter, we used the next available strip from a subsequent year.

To isolate drainage-related subsidence, we differenced the ArcticDEM strips within the 10-year lake mask and retained only pixels with negative elevation changes. Airborne radar observations across the GrIS suggest typical ice lid thicknesses of ~1.4 m and snow depths of ~0.65 m (Koenig et al., 2015), while modeling studies estimate total ice lid and snow thicknesses of 0.8 m- to 2.5 m under normal winter conditions (Law et al., 2020). To account for this, we subtracted a uniform 1.4 m from all DEM-derived depths before calculating lake volumes and mean depths. For three lakes that drained during winter 2023/2024, no post-drainage DEMs were available for that or subsequent seasons, and thus we were unable to calculate volumes and mean depths for these lakes.

#### 3.5 Modeling subglacial water pathways

To place winter lake drainage observations in the context of potential subglacial meltwater pathways, we calculated subglacial hydraulic potential gradients and associated flow routing (Shreve, 1972; Livingstone et al., 2013). Assuming subglacial water pressure equals the ice-overburden pressure (i.e., effective pressure N = 0), the hydraulic potential ( $\Phi$ ) is defined as:







$$\Phi = \rho_w g z_b + \rho_g z \tag{3}$$

where  $\rho_w$  is water density,  $\rho_i$  is ice density, g is gravitational acceleration,  $z_b$  is the bed elevation from BedMachine Version 5 (Morlighem et al., 2017; Morlighem, 2022), and z is the ice thickness. Because we are interested in routing pathways rather than the identification of subglacial lakes, all local sinks in the hydraulic potential surface are filled prior to flow modeling (Livingstone et al., 2013; Smith et al., 2017). Flow directions are then derived using the D8 algorithm (O'Callaghan & Mark, 1984), which routes flow downslope to one of eight neighboring cells. The resulting flow direction grid is used to generate an upstream water accumulation map, indicating potential subglacial drainage pathways beneath the ice sheet.

#### 3.5 Auxiliary data

To provide context about the intensity and spatial extent of surface melt during each melt season, we used daily ASCAT C-band normalized radar cross-section sigma naught ( $\sigma^0$ ) data to identify the presence or absence of melt across the 79NG and ZI basins. We used a threshold of -3.0 dB below / above the previous winter mean backscatter to define the presence / absence of melt, consistent with methods developed for C-band radar operating at 5.3 GHz (Ashcraft and Long, 2005). Grid cells were classified as experiencing melt on days when  $\sigma^0$  dropped below this threshold, which indicates the presence of wet snow relative to dry snow conditions.

This analysis was conducted for the 2014 to 2022 melt seasons; 2023 data were unavailable at the time of analysis. Winter backscatter means were calculated from data spanning 1 January to 31 March, while melt seasons were defined as 1 June to 31 August. The resulting daily binary melt / no melt maps were used to compute two metrics for each summer: (1) melt intensity for each grid cell, defined as the number of melt days; and (2) melt extent, defined as the total area experiencing melt. To enable comparison of melt intensity and extent between the years, we also calculated an annual melt index, defined as the sum of daily melt extents over the summer. We also used the daily binary melt / no melt data to calculate time series of melt extents (total area experiencing melt) across the entire region each summer.

To investigate possible dynamic effects of winter lake drainages on ice flow, we analyzed monthly ice velocity data (MEaSUREs Version 5; Joughin, 2023) from June 2015 to May 2024 along transects extending from near the termini to the upstream regions of both 79NG and ZI (Fig. 1b). For higher temporal resolution around suspected cascading lake drainage events, we also analyzed 6-and 12-day velocity data (MEaSUREs Version 2; Joughin, 2022) along the transects, focusing on periods before, during and after each event in the vicinity of affected lakes.

# 4 Results

#### 4.1 Winter lake drainage at Nioghalvfjerdsbræ and Zachariæ Isstrøm

Between the 2014/2015 and 2023/2024 winter seasons, a total of 90 lake drainage events were detected across 55 individual lakes in the 79NG (48 events) and ZI (42 events) basins (Fig. 3). Drained lakes ranged in elevation from a minimum of 88 m to a maximum of 1,046 m, with a mean of 485 m. Notably, only one drainage event was observed above 1000 m, despite 140 of the 404 lakes identified in the 10-year optical melt-season lake mask occurring above this elevation, and prior reports documenting lakes up to 1,500 m in this region (Schröder et al., 2020; Hochreuther et al., 2021; Lu et al., 2021) (Fig. 3).

Among the 55 draining lakes, 34 drained once over the ten winters, 12 drained twice, six drained three times, two drained four times, and one lake drained six times. The highest number of drainage events in a single winter was 18, recorded during 2018/2019.



In contrast, the fewest events - only four - occurred during both the 2020/2021 and 2021/2022 seasons (Fig. 4a). An increasing trend in drainage frequency is evident leading up to the 2018/2019 season, followed by a marked decline thereafter (Fig. 4a).

Drainage events were most frequent in September, accounting for approximately one-third of the total (31/90) (Fig. 4b). October and November had the second- and third-highest counts, respectively. While drainage frequency generally declined between September and February, a notable resurgence occurred in March and April, with nine and eight events respectively. Seven of the eight April events were linked to a single cascading drainage episode during the 2018/2019 season (see Section 4.3). Overall, drainage activity tended to decrease as each winter progressed, with drainages picking up again in late winter.

The three winters with the highest number of drainages - 2016/2017 through 2018/2019 - exhibit distinct temporal patterns in drainage timing (Fig. 4c). In 2016/2017, all drainages occurred by mid-January, the approximate midpoint of winter. In that year, there was a cluster of drainages in September followed by longer intervals between subsequent drainages. In contrast, drainages during the 2017/2018 season were more evenly distributed throughout the winter months. The 2018/2019 season featured drainages in both early and late winter, but these were more temporally clustered compared to 2017/2018. In the three winters that followed - 2019/2020 through 2021/2022 - drainage activity was markedly reduced. In both the 2020/2021 and 2021/2022 seasons, all drainages occurred prior to mid-December, indicating a shift toward earlier and fewer winter drainages during these years (Fig. 4c).

Although most draining lakes drained just once per winter, there were three cases where individual lakes drained twice within the same season - one each during 2014/2015, 2017/2018, and 2019/2020. All three lakes were located at low elevations (< 150 m), with two situated near the terminus of 79NG and one near the terminus of ZI. In each case, the second drainage occurred at least three months after the first, with initial events taking place before December and subsequent drainages occurring after the start of February.

Using timestamped ArcticDEM strips, we estimated the volumes and mean depths of 87 of the 90 observed winter lake drainages (Fig. 5). Data were unavailable for the three events that occurred during the 2023/2024 season. Across the 87 lakes, the mean depth was 4.10 m. The vast majority (85 of 87) had estimated volumes below ~0.02 km³. Two outliers, with drainage volumes of 0.052 ± 0.0096 km³ and 0.038 ± 0.0008 km³, were attributed to two separate drainages of the same lake, located at an elevation of 259 m near the ZI terminus (highlighted in Fig. 3). Excluding these two outliers, the mean lake drainage volume was 0.0039 ± 2.8 x 10<sup>-5</sup> km³.

**Figure 3.** Supraglacial lake locations from the 10-year melt season lake mask, alongside the locations and frequencies of winter drainage events observed between the 2014/2015 and 2023/2024 seasons. A black "+" marks a lake near the Zachariæ Isstrøm terminus that produced two exceptionally large drainage volumes (0.052 km³ and 0.038 km³). Dashed rectangles indicate the subregions shown in detail in Figs. 3, 9 and 10. Gray shading indicates land areas, and blue represents the ocean, based on the GIMP land-ice-ocean classification dataset (Howat, 2017).

**Figure 4.** Temporal patterns of winter lake drainage events over ten seasons (2014/15–2023/24). (a) Total number of drainages per winter. (b) Monthly distribution of drainage events from September to May, aggregated across all winters. (c) Timing, mean elevation and estimated volume of each drainage event, shown for each winter season. In (a) and (b), drainages from Nioghalvfjerdsbræ (79NG) and Zachariæ Isstrøm (ZI) are distinguished using dark grey and light grey bars respectively.

# 4.2 ASCAT melt season variability

The nine melt seasons preceding the 2014/2015 to 2022/2023 winters exhibited considerable variability in melt intensity (Appendix Fig. A1), melt extent (Fig. 6a, Appendix Fig. A2), and melt index (Fig. 6b). From 2015 to 2018, the melt index declined markedly from 1.88 × 10<sup>6</sup> km<sup>2</sup>-days to 4.61 × 10<sup>5</sup> km<sup>2</sup>-days (Fig. 6b), driven by concurrent decreases in both average melt intensity and melt extent (Appendix Figs. A1 and A2). This trend reversed in 2019 and 2020, with melt indices rising to 1.33 × 10<sup>6</sup> km<sup>2</sup>-days and 1.41 × 10<sup>6</sup> km<sup>2</sup>-days respectively. Notably, the winters of 2017/2018 and 2018/2019, which were among those with the highest number of lake drainages - 13 and 18 respectively (Fig. 4a) - were preceded by melt seasons with the two lowest melt indices (Fig. 6b). Conversely, the 2019/2020 and 2020/2021 winters, during which just six and four drainages occurred respectively (Fig. 4a), were preceded by summers with high melt intensities and extents (Fig. 6a). A closer comparison of the 2017 to 2020 melt extent time series shows that melt extent was typically lower in 2017 and 2018 than in 2019 and 2020 (Fig. 6a). The 2019 and 2020 time series also show both greater peak melt extents and more prolonged periods with melt extents exceeding 20,000 km<sup>2</sup>.

**Figure 5.** Histograms showing geometric characteristics of winter-draining lakes. (a) Distribution of estimated lake volumes. (b) Distribution of mean lake depths. Both metrics are derived from ArcticDEM timestamped strips for 87 winter-draining lakes.

**Figure 6.** (a) Daily melt extent time series from 1 June to 31 August for the 2017–2020 melt seasons, derived from ASCAT data. (b) Seasonal melt index from 2014 to 2022, calculated as the cumulative daily melt area during each summer melt season (June–August) based on ASCAT-derived surface melt extents.






## 4.3 Comparison of winter and summer lake drainage occurrence

To compare rapid lake drainages between winter and summer, we subset our dataset to include just the seven winter seasons from 2016/2017 to 2022/2023 and their preceding melt seasons (June to August, 2016 to 2022), for which summer drainage data over Northeast Greenland are available from Lutz et al., (2025) (Fig. 7, Appendix A3). During this period, 306 summer and 68 winter drainage events were recorded, yielding a summer-to-winter drainage ratio of approximately 4.5:1. These involved 129 unique lakes in summer and 42 in winter, corresponding to a summer-to-winter lake ratio of just over 3:1. On average, summer-draining lakes occurred at slightly lower elevations (458 m) than winter draining lakes (515 m) (Fig. 7).

The ratio of summer to winter drainages varied significantly across the years from ~1.3:1 in 2018/2019 to ~11.5:1 in 2019/2020. The 2018 melt season was notably cool, with a low melt index (Fig. 6b) and the fewest drainages (25) of any melt season. The subsequent 2018/2019 winter had the greatest number of drainages (18) of any winter. In contrast, the 2019 melt season was unusually warm, with a relatively high melt index (Fig. 6b), high melt intensities extending to higher elevations (~1500 m) than in other years (Appendix Fig. A2), and the greatest number of summer drainages (71). The subsequent 2019/2020 winter had just six drainages. From 2016 to 2018, the number of summer drainage events declined, while winter drainage counts remained relatively stable (Fig. 7a). After 2018, this pattern reversed: summer drainages increased, while winter drainages became less frequent.

Although many lakes exhibited repeated rapid drainage across multiple melt seasons (Lutz et al., 2025), and several showed recurring winter drainages in our study, drainage of the same lake in both a melt season and the subsequent winter was exceptionally rare. Only two such cases were identified: one during the 2016/2017 winter and another in 2018/2019. Similarly, most lakes that drained in winter did not drain again in the subsequent summer, with only 13 such instances observed: once in 2016/2017, 2018/2019, and 2020/2021; twice in 2019/2020 and 2021/2022; and six times in 2018/2019.

Several lakes exhibited drainage across multiple summer and winter seasons, displaying a range of behaviours. The most common pattern, observed in 39 lakes, involved a single drainage during one melt season with no winter drainage (Fig. 7b). Another 21 lakes drained once during winter and between zero and three times during separate melt seasons across the seven-year period. Only six lakes drained repeatedly across multiple summer and winter seasons (two or three drainage events in each). Two lakes drained a total of six times - across four winters and two summers (Fig. 7b).

To investigate relationships between winter drainage frequency, summer drainage frequency, and surface melt intensity / extent (as represented by the ASCAT melt index), we conducted a least-squares regression analysis (Fig. 8). The analysis spans seven summer-winter season pairs (df = 6), with each winter paired to the preceding summer. We report the correlation coefficient (r), coefficient of determination ( $R^2$ ), and slope (s), and consider  $p \le 0.1$  as the threshold for statistical significance due to the small sample size. Results are summarized in Table 2. Across the 25th, 50th, and 75th elevation quartiles, the number of winter drainages was negatively correlated with the number of preceding summer drainages (Fig. 8a), although this correlation was not statistically significant for the lowest quartile (p = 0.27). In contrast, high-elevation lakes in the fourth quartile exhibited a significant positive correlation (p = 0.085).

The summer-to-winter drainage ratio was significantly negatively correlated with the melt index (r = -0.83, p = 0.02) (Fig. 8b), indicating that as summer melt intensity and/or extent increases, the proportion of lake drainages occurring in winter decreases. Both the number of summer drainages and the total number of annual drainages were significantly positively correlated with the

melt index (p = 0.001 and p = 0.02, respectively) (Fig. 8c). In contrast, the number of winter drainages was negatively correlated with the melt index, although this trend was not statistically significant (p = 0.11) (Fig. 8c).

**Figure 7.** (a) Number of rapid lake drainages during the melt season (June-August; red bars) and subsequent winter (September-May; blue bars) from 2016/17 to 2022/23. Melt season drainage data are from Lutz et al. (2025). Hollow circles indicate the elevation of individual draining lakes each season of each year, while triangles represent the mean elevation. (b) Relationship between melt season and winter drainage frequencies for individual lakes. Green circle size represents the number of lakes sharing a specific combination of summer and winter drainage counts; numbers within circles indicate lake count.

**Figure 8.** Linear regression plots showing relationships between lake drainage metrics and the summer melt index for years 2016/2017 to 2022/2023. Solid lines indicate statistically significant correlations (p ≤ 0.1); dashed lines denote non-significant trends. (a) Number of winter drainages (September–May) versus number of drainages in the preceding summer (June-August), grouped by lake elevation quartiles. (b) Ratio of summer to winter drainages versus melt index. (c) Number of winter, summer, and total annual drainages (combined winter and preceding summer) versus melt index.


Table 2. Linear regression results corresponding to drainage frequency and melt index relationships shown in Fig. 8. Regressions are based on seven paired seasons (df = 6), where each winter season (September–May) is paired with the preceding summer (June–August). Reporting statistics include the coefficient of determination (R²), Pearson's correlation coefficient (r), slope (s), and p-value. Rows 1–4 show statistics for winter versus summer drainages by elevation quartile (Q1–Q4). Subsequent rows show statistics for the summer to winter drainage ratio versus melt index, summer drainages versus melt index, winter drainages versus melt index, and total annual drainages (winter plus preceding summer) versus melt index.

| Regression pair                             | $R^2$ | r     | S                        | p     |
|---------------------------------------------|-------|-------|--------------------------|-------|
| Winter vs. summer drainages Q1 (71–261 m)   | 0.20  | -0.44 | -0.21                    | 0.27  |
| Winter vs. summer drainages Q2 (262-393 m)  | 0.43  | -0.65 | -0.33                    | 0.08  |
| Winter vs. summer drainages Q3 (394–655 m)  | 0.40  | -0.63 | -0.30                    | 0.09  |
| Winter vs. summer drainages Q4 (656–1046 m) | 0.41  | 0.64  | 0.07                     | 0.09  |
| Summer-to-winter ratio vs. melt index       | 0.70  | 0.84  | $1.18 \times 10^{-5}$    | 0.02  |
| Summer drainages vs. melt index             | 0.90  | 0.95  | $4.67 \times 10^{-5}$    | 0.001 |
| Winter drainages vs. melt index             | 0.42  | -0.65 | -1.03 × 10 <sup>-5</sup> | 0.11  |
| Total drainages vs. melt index              | 0.71  | 0.84  | $3.65 \times 10^{-5}$    | 0.02  |

# 4.4 Cascading winter drainage events

We analyzed the timing and proximity of neighboring lake drainages alongside the maps of subglacial hydraulic potential and water pathways to identify lakes that may have been triggered by neighboring events via either stress coupling through the ice or hydrologic coupling along the subglacial drainage pathway. Following Christoffersen et al. (2018), we refer to these as *cascading drainage events*. Our analysis reveals that 46 of the 90 lake drainages observed over the ten winter seasons may have been part of such cascades. Of these, 18 drainages occurred across three major events involving five to seven lakes each, with the furthest lakes separated by ~25-33 km. The remaining 28 drainages were part of 12 smaller clusters, each involving two to three lakes in close proximity (<10 km). Thus, large cluster cascade events were less common than events involving small clusters. Below we describe the three large cluster events followed by three examples of smaller-scale clusters.

#### 4.4.1 Large cluster cascade events

Event 1: ZI, Winter 2016/2017. Between 5 September 08:43 and 6 September 17:32, five lakes drained on ZI (Fig.  $9a_i$ ). Lake elevations ranged from  $\sim$ 795 m (L1A) to  $\sim$ 400 m (L1E). Inter-lake distances were:

• L1A-L1B: 8 km

• L1B-L1C: 4.8 km

• L1C-L1D: 12 km

• L1D-L1E: 10.5 km

The total separation between L1A and L1E was ~33 km. All five lakes were connected via a modeled subglacial meltwater pathway. Nearby lakes at similar distances but lacking hydrologic connections did not drain.

Event 2: 79NG, Winter 2018/2019 Between 23 September 08:58 and 27 September 08:26, six lakes drained on 79NG (Fig. 9b<sub>i</sub>):

- L2A and L2B (~740–660 m) drained within ~36 hours and were 4.5 km apart.
  - L2C and L2D (~410–330 m) drained ~14 km downstream within the next 15 hours.

• L2E and L2F (~130 m) drained ~6 km further downstream over the following 48 hours.

Event 3: ZI, Winter 2018/2019. Between 17 and 20 April, seven lakes drained on ZI (Fig. 9c<sub>i</sub>):

- L3A and L3B (~400 m, ~1 km apart) drained between 17 April 18:29 and 19 April 08:26.
- L3C-L3F (~330 m, ~7 km downstream) drained within a 10-hour window.
- L3G (~630 m), located ~12 km upstream of L3A/B and ~19 km upstream of L3C-F, drained between 19 April 18:12 and 20 April 08:18.

#### 4.4.2 Small cluster cascade events

Event 4: 79NG, Winter 2017/2018. Between 16 and 18 September, lakes L4A and L4B (~350 m, ~1 km apart) drained within ~48 hours (Fig. 10a<sub>i</sub>). On 19 September, L4C (~166 m), ~8 km downstream, drained within ~24 hours.

Event 5: 79NG, Winter 2021/2022. Between 24 and 26 September, lakes L5A ( $\sim$ 751 m) and L5B ( $\sim$ 569 m),  $\sim$ 10 km apart, drained within  $\sim$ 48 hours (Fig. 10b<sub>i</sub>). This represents the largest separation observed among small-cluster cascade events.

Event 6: ZI, Winter 2022/2023. Between 15 and 16 March, three lakes (L6A–L6C) at  $\sim$ 260 m elevation drained within a  $\sim$ 24-hour period (Fig.  $10c_i$ ).

**Figure 9.** Location maps (i) and surface ice velocities (ii) for the three 'large cluster' cascading drainage events involving five to seven lakes. Panels (a<sub>i</sub>), (b<sub>i</sub>), and (c<sub>i</sub>) show the spatial distribution and drainage sequence of the cascades: (a<sub>i</sub>) L1A - L1E, occurring between 5 September 08:43 and 6 September 17:32, 2015; (b<sub>i</sub>) L2A - L2F, occurring between 23 September 08:58 and 27 September 2018; and (c<sub>i</sub>) L3A - L3G, occurring between 17 April 18:29 and 20 April 08:18, 2019. Drainage sequence during each event is color-coded from yellow (first) to red (third). Drained lake polygons are overlaid on the modelled subglacial hydraulic potential grid, with subglacial flow paths shown as lines shaded from grey to black, indicating increasing upstream area accumulation. Black transect lines, going from higher to lower elevations, closely follow lake locations. Velocity plots (ii) correspond to each lake cascade and show ice surface velocities sampled at 200 m intervals along the transects. Data are from 6-and 12-day MEaSUREs Version 2 SAR image pairs; date ranges for each pair are shown in the legends. Approximate lake locations are marked with blue triangles.

Figure 10. Location maps (i) and surface ice velocities (ii) for the three cascading drainage events involving between two and three lakes. Panels (a<sub>i</sub>), (b<sub>i</sub>), and (c<sub>i</sub>) show the spatial distribution and drainage sequence of the cascades: (a<sub>i</sub>) L4A – L4E, occurring between 16 September 08:58 and 18 September 08:42, 2017; (b<sub>i</sub>) L5A – L5B, occurring between 24 September 18:05 and 26 September 17:49, 2021; and (c<sub>i</sub>) L6A – L6C, occurring between 17 April 18:29 and 20 April 08:18, 2019. Drainage sequence during each event is color-coded from yellow (first) to orange (last). Drained lake polygons are overlaid on the modelled subglacial hydraulic potential grid, with subglacial flow paths shown as lines shaded from grey to black, indicating increasing upstream area accumulation. Black transect lines, going from higher to lower elevations, closely follow lake locations. Velocity plots (ii) correspond to each lake cascade and show ice surface velocities sampled at 200 m intervals along the transects. Data are from 6-and 12-day MEaSUREs Version 2 SAR image pairs; date ranges for each pair are shown in the legends. Approximate lake locations are marked with blue triangles.

# 4.5 Ice velocities



Monthly ice velocity time series from 2015 to 2024 for transects along 79NG and ZI are shown in Fig. 11. Near-terminus velocities are higher at ZI, ranging from ~1600 to 2200 m yr<sup>-1</sup> at ~5 km inland, and from ~1200 to 1600 m yr<sup>-1</sup> at ~10 km. In contrast, velocities at 79NG are slower, ranging from ~1000 to 1250 m yr<sup>-1</sup> at both 5 km and 10 km from the terminus. At ZI, seasonal velocity variations are pronounced. Velocities typically peak in July, then decrease sharply through to September or October, often dropping to pre-melt season values to reach the annual minimum. Velocities then increase gradually throughout the winter. An exception to this pattern occurred during the anomalously cool 2018 melt season, when the late-summer slowdown was minimal






and winter velocities remained relatively constant. At 79NG, seasonal velocity cycles are more subdued compared to ZI, showing weaker summer peaks and less pronounced winter acceleration.

Throughout the study period, 79NG and ZI exhibited distinct interannual velocity trends. ZI showed a general increase in velocities, particularly towards the terminus, while velocities at 79NG remained relatively stable. At the monthly scale, no obvious increases in ice velocity were observed that could be attributed to winter drainage events. For both glaciers, winters with the highest number of drainages (2016/2017 and 2018/2019) showed velocity patterns comparable to other winters, including those with few drainage events (2019/2020 to 2021/2022) (Fig. 4a and 11).

The 6- and 12-day SAR-derived MEaSUREs velocities were sampled every 200 m along transects tracking both large cluster cascades (involving five to seven lakes; Events 1-3; Fig. 9a; to c;) and smaller cluster events (of two or three lakes; Events 4-6; Fig. 10a; to 10c;), as outlined in Section 4.4. During Event 1 (5 September 08:43 to 6 September 17:32, 2015), data coverage was limited, and no velocity changes associated with the event were detected (Fig. 9a;). Event 2 (23 September 08:58 to 27 September 08:26, 2018) had good velocity coverage (Fig. 9b;). Elevated velocities were observed along the transect between 18 and 23 September, particularly between 0-15 km and, to a lesser extent, from 25-35 km. These elevated velocities appear temporally associated with the drainages of L2A and L2B (23 September 08:58 to 24 September 17:47), although a definitive link cannot be made due to the timing of the 6-day image pairs (18 - 23 September and 24 - 29 September). Elsewhere along the transect, velocities remained largely unchanged before, during and after the drainage of the remaining lakes of the cascade. Event 3 (17 April 18:29 to 20 April 08:18, 2019) appears to be generally associated with a velocity increase, captured by the 16 - 21 April image pair (Fig. 9c;). Around 3 km along the transect, near L3G, a localised velocity spike occurred during the drainage, followed by a return to pre-event levels. From ~10 km onward, velocity increases were more pronounced, exceeding pre drainage ones by ~40 -100 m yr <sup>1</sup>. A particularly prominent velocity spike occurred ~15 km, directly adjacent to L3A and L3B. Additional changes included a sharp velocity increase at ~26 km and a marked decrease at ~27 km, near L3C to L3E and lake L3F, respectively (Fig. 9c;). Following the cascade, ice velocities generally returned to pre-drainage values.

During Events 4-6 (Fig.  $10a_i$  to 10c), there is no consistent or clear ice velocity response observed before, during or after the lake drainages, likely due in part to gaps in the velocity data. For Event 4, only a single, isolated velocity spike is observed ~11 km along the transect, adjacent to L4C (Fig.  $10a_{ii}$ ). Coverage is limited for Event 5, and no definitive velocity anomalies are detected, although a localised spike followed by sudden drop near 12 km - coinciding with L5B - may suggest a short-lived response prior to its drainage (Fig.  $10b_{ii}$ ). In Event 6, a general decrease in velocity is observed along the transect, with a pronounced drop at ~9 km (Fig.  $10c_{ii}$ ).

**Figure 11.** Monthly ice surface velocity time series from June 2015 to January 2024 for Nioghalvfjerdsbræ (79NG) and Zachariæ Isstrøm (ZI), derived from monthly MEaSUREs data. Velocity measurements correspond to points along the transects shown in Fig. 1b.

# 5 Discussion




# 525 5.1 Implications of winter lake drainage at Nioghalvfjerdsbræ and Zachariæ

To date, relatively few studies have documented winter lake drainages on the GrIS (Schröder et al., 2020; Benedek & Willis, 2021; Maier et al., 2023). Our study, which spans 10 consecutive winters and uses SAR imagery with temporal resolutions ranging from twice daily to 12 days, provides the first evidence that winter drainage events are not only occurring but are quite common. These findings highlight winter drainage as an important and previously underrecognized fate for supraglacial lakes—alongside summer drainage (Das et al., 2008; Selmes et al., 2011; Tedesco et al., 2013), freeze-over, burial by snow with overwinter survival (Koenig et al., 2015; Lampkin et al., 2020; Law et al., 2020), and complete winter freeze-through (Miles et al., 2017; Hossain et al., 2024).

Over the last 40 years, supraglacial lakes have been forming (Howat et al, 2013) and draining (Otto et al, 2022) at progressively higher elevations during the summer, a trend projected to continue under future climate scenarios, with some of the greatest rates of expansion in NE Greenland (Leeson et al., 2015; Ignéczi et al., 2016, Fan et al., 2025). This inland expansion of lakes could increase the likelihood of surface-to-bed meltwater transfer in regions that are currently considered relatively stable (Doyle et al., 2014). This would have implications for both subglacial hydrology and ice dynamics (Mayaud et al., 2014), although the likelihood of this has been questioned (Poinar et al., 2015). Our 10-year record indicates that winter drainage events remain largely confined to lower elevations (

- Our analysis of melt season conditions, using ASCAT-derived melt products and the summer lake drainage dataset from Lutz et al., (2025), suggests that summer conditions influence the likelihood of winter lake drainage. Specifically, winters following summers with a high melt index and more frequent summer drainage events tend to exhibit fewer winter drainages (Fig. 8; Table 2). Conversely, winters following summers with a lower melt index are associated with a greater number of winter drainage, at all but the highest elevations (Fig. 8 and Table 2).
- Two interrelated mechanisms may explain this pattern. First, during high-intensity melt seasons, enhanced surface-to-bed meltwater inputs can elevate basal water pressures, increase ice flow velocities, and induce greater stress perturbations conditions that promote widespread summer lake drainage. This likely results in fewer lakes persisting into winter, thereby reducing the probability of winter drainage events. This explanation is supported by the observed positive correlation between the melt index and summer drainages, and the corresponding negative correlation with winter drainages (Fig. 8; Table 2).
- Second, in low intensity melt seasons, the subglacial drainage system is less likely to evolve into an efficient, channelized network (Banwell et al., 2016; Andrews et al., 2018). As a result, the bed remains more hydraulically inefficient and sensitive to any additional meltwater inputs. For example, following the relatively cool summer of 2018 and into the early winter of 2018/19, the subglacial hydrological system was likely particularly inefficient, creating favourable conditions for winter drainage activity.
- Two of the three large cluster cascade events (Events 2 & 3) identified in our ten-year record occurred during the 2018/2019 winter.

  Notably, Event 3 was the only late-winter cascade observed in the dataset a seven-lake drainage event between 17 and 20 April 2019 (Fig. 9c<sub>i</sub>). No other winter exhibited large cluster cascades during the late winter. In contrast, the 2017/2018 winter, which also followed a low-intensity melt season, did not feature any large cascade events. Instead, lake drainages during this winter were more evenly distributed throughout the season. However, 2017/2018 experienced three of the 12 small cluster cascades observed in this study the highest number recorded in a single winter.
- This difference may suggest the influence of cumulative subglacial hydrological memory. While both winters followed low-intensity melt seasons, the 2018/2019 winter was preceded by two consecutive low-melt summers, potentially leaving the subglacial drainage system in a more hydraulically inefficient and sensitive state. In contrast, 2017/2018 followed only a single low-melt summer, which may have resulted in less widespread system inefficiency. This highlights the potential importance of multi-year melt history in conditioning subglacial responses and the likelihood of cascading winter lake drainages.
- Surface-to-bed water transfer from winter lake drainages may also influence ice sheet stability near the grounding lines of Greenland's marine-terminating glaciers, similar to the effects of melt-season surface inputs that enter subglacial pathways and discharge at calving fronts (Rignot et al., 2010; How et al., 2017). Zeising et al., (2024) have shown that submarine meltwater inputs driven by summer lake drainages play a role in controlling basal channel growth beneath the floating ice tongue of 79NG the largest in Greenland. The occurrence of winter lake drainages introduces a potentially important, but previously overlooked, mechanism that could similarly influence the stability of ice tongues. This influence is likely to be most significant at low elevations and early in the winter, when subglacial channels are more likely to remain open, increasing the probability that meltwater reaches the terminus and drives basal melting near the grounding line.
  - Although our study does not quantify winter lake areas directly, the findings of Dunmire et al. (2021) and Zheng et al. (2023) provide useful context. In Northeast Greenland, Dunmire et al. (2021) reported that the extent of buried lakes during the 2018/19 winter was just 17% of that observed in 2019/20. Similarly, Zheng et al. (2023) found that winter lake area in 2018/2019 was only







~9% of the 2019/2020 extent. When compared with our results, these findings suggest that a greater winter lake extent does not necessarily correspond to a higher number of winter drainages. In fact, the winter with the most observed drainages (2018/19) coincided with the smaller of the two lake extents, indicating that lake presence and area alone are not reliable predictors of winter drainage frequency.

### 5.2 Hypothesized mechanisms of cascading winter lake drainages

Over the ten winters studied, cascading lake drainages account for approximately half of all events - comparable to summer drainage patterns, where lake drainage clusters are also common (Morriss et al., 2013; Fitzpatrick et al., 2014; Christoffersen et al., 2018; Hoffman et al., 2018). We identify two distinct modes of winter cascading behavior: (1) *small cluster cascades* involving two or three lakes separated by <10 km); and (2) *large cluster cascades* involving five to seven lakes extending ~25–33 km. Below, we examine each in detail, emphasizing the likely triggering mechanisms.

#### 5.2.1 Small cluster cascades

We propose that small cascades are initiated by rapid basal cavity opening or slip during the drainage of an initial lake, which generates horizontal tensile stresses sufficient to trigger subsequent nearby lake drainages. Similar mechanisms have been reported for summer cascades by Stevens et al., (2024). These stress perturbations, often lasting only a few hours (Das et al., 2008; Tedesco et al., 2012; Doyle et al., 2013; Stevens et al., 2015; Chudley et al., 2019; Lai et al., 2021), typically influence lakes within a few kilometers (Stevens et al., 2024).

Event 4 in September 2017/18 supports this interpretation (Fig. 10a<sub>i</sub>). Within ~48-hours (16 September 08:58 to 18 September 08:42), lakes L4A and L4B - ~1 km apart - both drained. This pairing occurred in four out of ten winters, mirroring Stevens at al., (2024), who found synchronous drainage of neighbouring lakes in ~50% of summers from 2000-2023. Where ice thickness and basal topography are favourable, rapid drainage of one lake may, through basal cavity opening, produce tensile stresses capable of initiating drainage of a neighboring lake (Stevens et al., 2024).

Lake L4C, ~8 km down glacier, did not drain concurrently. Instead, its drainage occurred ~24-hours later. Because the modeled subglacial drainage pathway beneath L4C follows a different branch than that beneath L4A and L4B, L4C likely drained indirectly - via stress gradients transmitted through the ice, as water flows downstream from L4A and L4B, rather than directly through the transmission of a high-pressure wave beneath L4C. Supporting this interpretation is a coincident velocity spike near L1C between 17 and 22 September (Fig.  $10a_{ii}$ ).

Event 6, between 16 and 17 March - involving lakes L6A to L6C, each < 3 km apart – is also consistent with rapid tensile-stress transmission from basal cavity opening (Fig. 10c<sub>i</sub>). Conversely, in Event 5, the 24-26 September drainage of lakes L5A and L5B, separated by  $\sim 10$  km, likely involved a different mechanism (Fig. 10b<sub>i</sub>). Instead, the near-synchronous drainages (within  $\sim 48$  hours) may reflect: (1) subglacial water transfer from L5A to L5B causing basal uplift and/or slip if L5A drained first; or (2) downstream ice acceleration affecting upstream stress fields if L5B drained first.






## 5.2.2 Large cluster cascades

The second, less frequent mode involves five to seven lakes over distances of ~25-33 km distances. These larger cascades were observed three times - September 2016, September 2018, and April 2019 - likely resulting from a combination of hydrological and ice-dynamic mechanisms.

During Event 2 (Fig. 9bi), drainage began with lakes L2A and L2B (~5 km apart), which emptied within a 36-hour period. Approximately 15-hours later L2C and L2D (~14 km downstream) drained, followed by L2E and L2F (~6 km farther downstream) within the next ~16-hours. Because L2A and L2B lie along the same subglacial pathway are too far apart for basal-cavity-opening tensile stresses alone to explain the sequence (Stevens et al, 2024), we suggest that subglacial water movement and associated basal uplift/slip propagated drainage downstream. Similar processes likely triggered the drainage of L2C to L2F. Elastic stresses from rapid drainage persist for only a few hours (Doyle et al., 2013; Chudley et al., 2019; Lai et al., 2021; Stevens et al., 2024), making rapid sequential triggering plausible. Meltwater from the drainage of L2C or L2D may have contributed to rapid basal cavity opening, helping to trigger the drainage of the other lake, if the two did not drain through the same fracture. The same may be true of lakes L2E and L2F.

Event 3, the April 2019 cascade also likely involved multiple lake drainage mechanisms. L3A and L3B drained between 17 April 18:29 and 19 April 08:26, with L3C to L3F, located ~8 km downstream along the modeled subglacial pathway, draining within the following 10 hours (Fig. 9c<sub>i</sub>). The proximity of L3A or L3B (<1 km) suggests that drainage of one may have triggered the other via tensional stresses caused by basal cavity opening. The later drainage of L3C to L3F, located farther along the subglacial pathway, was likely triggered by basal uplift and/or slip resulting from the subglacial transit of meltwater from L3A and L3B. As in the September 2018 cascade, the proximity of L3C to L3E suggests that tensile stresses from cavity uplift during one lake's drainage could have contributed to triggering adjacent lakes. L3G drained within ~24-hour hours of the L3C-L3F sequence (Fig. 9c<sub>i</sub>). Although L3G is hydrologically connected via the subglacial pathway, it lies ~11 km upstream, making it unlikely that a pressure wave travelled that distance to induce uplift or slip beneath it. Instead, the rapid drainage of L3A-F likely caused localized ice acceleration, placing the L3G basin under horizontal tension and enabling hydrofracture-driven drainage (Christoffersen et al., 2018).

Event 1, between 5 September 08:43 and 6 September 17:32, 2015, involved the drainage of lakes L1A to L1E (Fig. 9a<sub>i</sub>). This cascade involved lakes more evenly spaced than those in the 2018/2019 cascades, with greater separations ranging from ~3.5 km to ~32 km. Although the exact sequence is unknown, it is likely that drainage was driven primarily by meltwater flow along the shared subglacial pathway. L1B and L1C were close enough for basal cavity opening in one to potentially trigger the other, while the wider spacing of the remaining lakes makes this less likely. If L1A drained first, subglacial meltwater would travel along the subglacial pathway, triggering downstream lakes to drain via uplift and/or slip. Alternatively, drainage of a downstream lake could have transmitted stress gradients upstream through the ice, triggering the upstream lakes through horizontal tension.

In summary, smaller drainage cascades involving two or three lakes are most likely driven by a single triggering mechanism. In contrast, larger cascades, involving five to seven lakes, require a combination of mechanisms, including: (1) tensile stress from basal cavity opening during rapid drainage of a nearby lake; (2) basal uplift and/or slip caused by subglacial meltwater passage from the drainage of an upstream lake; (3) horizontal tension induced by downstream lake drainage and ice acceleration. While these mechanisms can explain lake-to-lake triggering during a cascade, they do not account for the initial drainage trigger or




isolated winter events. These may be initiated by transient strain rate anomalies within the ice (Poinar and Andrews, 2021) or episodic subglacial drainage that induces vertical uplift and basal acceleration (Andersen et al., 2023).

#### 5.3 Ice velocity responses

Large cluster cascades (involving five to seven lakes) that occurred in early winter (September) caused only limited ice acceleration (Fig. 9a<sub>ii</sub> and 9b<sub>ii</sub>), especially when compared to the late winter (April) cascade (Fig. 9c<sub>ii</sub>). Because these events took place near the end of the melt season, it is likely that an efficient subglacial system existed, limiting the potential for lake drainages to generate high basal water pressures and reduce basal friction (Tedstone et al., 2013; Andrews et al., 2014; Andrews et al., 2018). This interpretation aligns with observations of seasonal slowdowns following the close of most melt seasons (Fig. 11).

Similarly, small cluster cascades (involving two or three lakes) that occurred in early winter also had minimal or no discernible impact on ice velocity (Fig. 10aii and 10bii). Although the 6- and 12-day ice velocity products provide relatively high temporal resolution, short-lived velocity perturbations associated with lake drainage clusters may go undetected (Poinar & Andrews, 2021; Stevens et al., 2024). However, in-situ measurements corroborate our findings: at Helheim Glacier, GPS-derived velocities remained largely unaffected by a near-terminus lake drainage event in the late melt season, likely due to the presence of an efficient subglacial drainage system (Stevens et al., 2022). Similarly, a September drainage event near the terminus of 79NG produced only a minor and short-lived velocity increase (Neckel et al., 2020).

The late winter large cluster cascade produced a significant increase in ice velocity (Fig. 9c<sub>ii</sub>). Yet even this response was short-lived, with ice velocities returning to predrainage velocities within the subsequent 6-day interval. This suggests that, although the water input was substantial and likely entered an inefficient subglacial system, it was still rapidly evacuated - likely through pre-existing or transient subglacial pathways from a basal blister beneath the lake. Considering the large cluster that occurred in early winter 2018 (Fig. 9b<sub>ii</sub>), the timing of the drainage relative to the 6-day ice-velocity image pairs makes it difficult to determine whether the event was fully captured. A possible explanation for the limited velocity response observed between 25 and 33 km along the transect is that the subglacial system in this region was already more efficient due to its proximity to the terminus. Zeising et al. (2024) reported the presence of large basal channels near the grounding line of 79NG, consistent with the location of the large cluster drainage.

Overall, we find no evidence that the number of winter lake drainages correlates with changes in monthly winter ice velocities at 79NG and ZI (Fig. 11). This suggests that the volume and rate of meltwater reaching the bed during these events are insufficient to sustain elevated subglacial water pressure or reduce basal friction over extended periods. The ongoing year-to-year winter acceleration observed at ZI is instead attributed to other factors, most notably its rapid retreat beginning in 2012 (Mouginot et al., 2015; Grinsted et al., 2022; Khan et al., 2022).

Maier et al., (2023) reported a 15-lake cascade in a land-terminating region of southwest Greenland, identified using optical imagery. Using Differential Interferometry Synthetic Aperture Radar (DInSAR), they tracked the down-glacier propagation of a velocity wave associated with subglacial water movement, which lasted approximately a month. Together with our findings from the April 2019 cascade (Fig. 9c), this suggests that winter cascades can influence ice dynamics and potentially affect the evolution of subglacial drainage systems.







It is well established that during the melt season, increasing surface meltwater inputs - including lake drainages - initially enhance basal cavity expansion and glacier acceleration, followed by subglacial channel development and a transition to lower velocities (Bartholomew et al., 2010; Tedstone et al., 2013; Andrews et al., 2014; Banwell et al., 2016; Andrews et al., 2018). However, whether winter lake drainages can trigger similar subglacial system development remains an open question. Further investigation is needed to assess whether winter drainage events can precondition the subglacial drainage system in ways that influence ice dynamics at the onset of the subsequent melt season.

# 5.4 A new method for detecting winter supraglacial lake drainages

We developed a semi-automated method to detect winter lake drainage events at high temporal resolution using incidence-angle-normalized C-band SAR. This enabled observations of winter lake behaviour that had not previously been possible. Incorporating imagery from multiple sensor geometries allowed us to detect drainages with sub-daily temporal resolution - substantially improving the precision of drainage timing compared to the standard 6- and 12-day repeat cycles of Sentinel-1 or the 4-day cycle of RCM. This was particularly advantageous for capturing cascading drainage events involving multiple lakes over several days, where higher temporal resolution helped distinguish individual drainage sequences. In contrast, coarser revisit intervals can result in temporal aliasing, causing such cascades to appear as a synchronous event (Cooley & Christoffersen, 2017; Stevens et al., 2024). Focusing on the winter season - when surface meltwater production and lake filling are minimal - reduced the influence of slow drainage mechanisms such as overtopping and channel incision. This allowed us to isolate rapid drainage events, which are more likely to result from hydrofracture.

We tracked mean  $\overline{\sigma}^{\circ}_{HV35}$  of lakes during winter using a 10-year Landsat-derived summer lake mask, following the approach of Benedek & Willis (2021). This enabled the detection of drainage from lakes that remained buried for one or more years - events previously documented across Greenland, including at 79NG and ZI (Koenig et al., 2015; Lampkin et al., 2020). The use of this long-term mask also allowed for the inclusion of partially snow- or ice-covered lakes, many of which would have been missed using only single-year masks, e.g., those following the low-melt summer of 2018, which showed no visible surface expression (Dunmire et al., 2021; Zheng et al., 2023).

Employing a 10-year composite mask introduced large variability in the magnitude of  $\overline{\sigma}^{\circ}_{HV35}$  increases during drainage events. The degree to which a lake fills within its long-term extent depends on melt season intensity and basin morphology. Lakes that fill up only partially may exhibit higher pre-drainage  $\overline{\sigma}^{\circ}_{HV35}$  due to a lower meltwater-to-ice facies ratio, resulting in smaller relative backscatter increases during drainage events. Consequently, drainage of small lakes within the larger composite extent may go undetected. We consider this to be a rare occurrence, as smaller are less likely to persist into winter (Lampkin et al., 2020), and their shallower depths make them more prone to freeze through (Law et al., 2020). Lake elevation also influences the magnitude of  $\overline{\sigma}^{\circ}_{HV35}$  changes. Lower-elevation lakes typically overlay more crevassed or rough glacier surfaces, resulting in higher post-drainage backscatter  $\overline{\sigma}^{\circ}_{HV35}$  than those at higher elevations, where ice slabs or firn dominate.

Despite the strengths of our approach, fully-automating winter lake drainage detection remains challenging. Manual screening is still needed to filter false positives, particularly during early winter freeze-up, when widespread increases in backscatter occur across all glacier surface facies. Additionally, while our method effectively identifies rapid winter drainages and cascading events at high temporal resolution, it does not resolve the areal extent of buried lakes during the winter or freeze through, as in previous







studies (Miles et al., 2017; Schröder et al., 2020; Dunmire et al., 2021, Zheng et al., 2023). Thus, we are unable to quantify seasonal variations in stored meltwater or assess the completeness of individual drainage events.

#### 5.5 Future work

Winter surface lake drainage may significantly influence subglacial hydrology, with potential implications for ice dynamics. Our results show that winter lake drainages at ZI and 79NG are relatively common, comprising 22% of all observed drainages over a seven-year period. Unlike optical datasets, such as MODIS and Landsat, which provide multi-decadal archives, no SAR archive prior to the launch of Sentinel-1 in 2014 (and RCM in 2019) offers sufficient spatial and temporal coverage to identify historical winter drainage events. This highlights the need to extend winter lake drainage investigations to other regions of the Greenland Ice Sheet to evaluate their broader prevalence. Winter monitoring is especially critical as surface lakes are expected to migrate further inland with ongoing climate change.

While numerous studies have investigated lake behavior during melt seasons using optical imagery (e.g., Sundal et al., 2009; Selmes et al., 2011; Morriss et al., 2013; Fitzpatrick et al., 2014; Williamson et al., 2018; Fan et al., 2025; Lutz et al., 2025), recent research using satellite SAR and airborne radar has shown that surface lakes can persist through winter and into the following melt season (Koenig et al., 2015; Miles et al., 2017; Dunmire et al., 2021; Zheng et al., 2023). Furthermore, emerging evidence including our own findings - suggests that winter drainage is a plausible and perhaps common fate for surface lakes (Schröder et al., 2020; Benedek & Willis, 2021; Maier et al., 2023). For ZI and 79NG, we integrated our winter lake drainage data with summer drainage observations from Lutz et al (2025), offering one of the first continuous, year-round drainage records. Future studies should build on this approach by combining optical and SAR-based methods to support comprehensive monitoring of lake life cycles and their impacts on ice sheet dynamics.

Optical remote sensing remains highly effective for mapping lake extent, estimating water volume, and detecting summer drainage events (Pope et al., 2016; Williamson et al., 2018). In contrast, SAR uniquely enables detection of drainage beneath snow cover and during the polar night, as well as lake areal extent in winter (Miles et al., 2017; Benedek & Willis, 2021; Zheng et al., 2023). While low dielectric contrasts between meltwater-saturated snow and firn pose challenges, SAR still offers a valuable complement to optical observations during summer, particularly under persistent cloud cover (Schröder et al., 2020). The long-term C-band SAR record from Sentinel-1, now bolstered by the launch of Sentinel-1C and the planned Sentinel-1D, provides a solid foundation for advancing winter hydrological studies on the Greenland Ice Sheet and beyond.

Greater understanding is needed regarding how winter lake drainages affect ice dynamics. Summer studies show that the initial input of surface meltwater typically accelerates ice flow, followed by deceleration as subglacial drainage systems become more efficient (Sole et al., 2013; Tedstone et al., 2013; Andrews et al., 2014). However, the influence of winter drainage on subglacial channel development or on ice flow during the subsequent melt season remains unclear. Winter lake behavior could also influence supraglacial hydrology. For example, lakes that persist through the winter are more likely to contribute to meltwater routing via supraglacial river networks in summer (Lampkin et al., 2020). As such, winter drainage events may play a key role in shaping the spatial and temporal distribution of surface meltwater at the onset of the melt season.

Although satellite-derived ice velocity products with 6–12 day resolution are valuable, they are unlikely to capture the short-lived dynamic responses triggered by rapid winter drainage events (Poinar & Andrews, 2021; Stevens et al., 2024). Integrating SAR observations with high-frequency (






750 ice deformation, as has been demonstrated for melt season drainages (Das et al., 2008; Doyle et al., 2013; Dow et al., 2015; Chudley et al., 2019; Stevens et al., 2024). Extending these measurements into winter would help determine whether winter drainage processes are fundamentally different from those in summer and provide essential ground truth for interpreting satellite-based observations across the ice sheet.

#### **6 Conclusions**

We investigated ten years of winter supraglacial lake drainages on the Greenland Ice Sheet using C-band SAR data from Sentinel-1 and the RADARSAT Constellation Mission. Employing a semi-automated detection algorithm that tracks backscatter changes and incorporates a new incidence angle normalization method, we combined data from multiple viewing angles to achieve neardaily temporal resolution.

Our analysis reveals that winter lake drainages are a common occurrence at Zachariæ Isstrøm (ZI) and Nioghalvfjerdsbræ (79NG) in northeast Greenland. Over the study period, we identified 90 drainage events across 55 lakes, with some lakes draining up to six times in ten winters. Drainage frequency generally declined as winter progressed, though timing and location varied considerably.

Approximately half of all winter drainages occurred as part of cascading events involving two to seven lakes. We identified two distinct cascade types: (1) small-scale cascades involving two or three lakes over short distances (

# Appendix A

Figure A1. Maps of melt intensity (cumulative melt days between 1 June and 31 August) for Nioghalvfjerdsbræ (79NG) and Zachariæ Isstrøm (ZI) derived from ASCAT data.

**Figure A2.** Time series of ASCAT-derived melt area across the entire domain from 1 June to 31 August of the 2014 to 2022 melt seasons. Three of these time series are also shown in Fig. 6.

**Figure A3.** Locations of melt season (red polygon) and winter (blue polygon) rapid drainages from 2016 to 2022. Melt season rapid lake drainages are from Lutz et al., (2025). Gray shading indicates land areas, and black represents an ocean mask, both derived from the GIMP lake, ice, and ocean classification dataset (Howat, 2017).

# Data availability



Sentinel-1 C-band SAR GRD scenes are available from the Alaska Satellite Facility (ASF DAAC) [https://search.asf.alaska.edu/]. RADARSAT Constellation Mission (RCM) C-band SAR GRD scenes (2019–2024) were obtained from the Natural Resources Canada Earth Observation Data Management System (EODMS) [https://www.eodms-sgdot.nrcan-rncan.gc.ca/] and are accessible upon registration; redistribution of the raw scenes is restricted by the provider's licence.

Landsat 8/9 OLI Tier-1 L1TP images (2014–2023) were obtained from the USGS EarthExplorer [https://earthexplorer.usgs.gov/].

ArcticDEM v4.1 mosaics and strips used for terrain correction and lake-volume estimates are available from Harvard Dataverse (mosaics & strips: https://www.pgc.umn.edu/data/arcticdem/). BedMachine Greenland v5 is distributed by NSIDC (https://nsidc.org/data/idbmg4/versions/5). The MEaSUREs Greenland Ice Mapping Project (GIMP) land/ice-ocean classification mask is available from NSIDC (https://nsidc.org/data/nsidc-0714/versions/1). Glacier basin outlines used to delineate 79NG and ZI are available from Dryad (https://datadryad.org/dataset/doi:10.7280/D1WT11).

ASCAT C-band from [https://www.scp.byu.edu/]. Ice-velocity data are the MEaSUREs velocity mosaics (multi-year, 6/12-day, and monthly products) from NSIDC (multi-year mosaic https://nsidc.org/data/nsidc-0670/versions/1; 6/12-day https://nsidc.org/data/nsidc-0766/versions/2; monthly mosaic https://nsidc.org/data/nsidc-0731/versions/5).

Derived data products generated in this work (winter-drainage event lake polygons) are available from the corresponding author upon request.

#### **Author contribution**

C.D. led the conceptualization of the study, carried out the methodology, performed the formal analysis, and prepared the original draft. R.S. supported study conceptualization, methodology design, and manuscript editing. I.W. supported study conceptualization, methodology design and contributed to reviewing and editing. K.M. assisted with data processing and analysis of ASCAT datasets. R.S. acquired funding and provided supervision. All co-authors contributed to editing the manuscript.

#### 830 Competing interests

The authors declare that they have no conflict of interest.

# Financial support

This research has been supported by the University of Victoria graduate program. R.S. has been supported by the Natural Sciences and Engineering Research Council of Canada (NSERC) Discovery Grants program (110\_2022\_2023\_Q1\_2082).

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
