# Peer review of "A decade of winter supraglacial lake drainage across Northeast Greenland using C-band SAR"

_EGUsphere, 2025_

## Referee Comment (RC1)

**A decade of winter supraglacial lake drainage across Northeast Greenland using C-band SAR**

Authors: Connor Dean, Randall Scharien, Ian Willis, and Kali McDougall

Review by Katrina Lutz, November 2025

**General comments**

This manuscript presents a novel winter supraglacial lake drainage detection methodology using a combination of two SAR data sources. This advanced methodology normalizes for differences in acquisition geometry and satellite missions, allowing for a cohesive output with a high temporal resolution. A thorough literature review was conducted and presented in a logical manner, relating the works of other groups effectively to the topic analyzed in this manuscript. The methodology seems well thought-out and incorporates many data sources and processing chains to provide meaningful context to the drainages, in particular the volume estimation using ArcticDEM strips, subglacial water pathways mapping, and the integration of ASCAT data for surface melt estimation. The results are overall well presented and the integration of the summer drainage dataset with their winter dataset provides a novel insight into how the drainage behaviors in each season influence each other. The writing style is elevated yet easy to read; however, some restructuring and concision is needed for the final discussion sections and conclusion. Overall, I find this manuscript to be of a high quality in both methodology and presentation, with some minor work needed to clarify the details of some processes and bring the work to a succinct conclusion.

**Specific comments**

**L16:** "…exhibiting substantial interannual variability, ranging from a maximum…"

**L54**: Adjust the phrase "ice layers" to more clearly reference upper ice layers

**L70:** There are several instances of the British spelling of some words being used, while the majority of the manuscript is written with American English spelling. Please adjust the spellings so that they are consistent (specific examples include behaviour, analysed, favourable).

**L104-109:** An important fact about 79NG is that it has an extensive floating tongue. This fact should be mentioned in this paragraph and related to the study area you define in Fig. 1 (mentioning the fact that lakes that form on the floating tongue are not analyzed). The terminology used to describe your study area must also be changed to reflect the difference between the terminus and the grounding line. There are several instances throughout the manuscript where 79NG's terminus is referenced but the grounding line is meant – these two are not interchangeable for a glacier with a floating tongue and could lead to confusion by the reader.

**L109:** I believe where Fig. 2.1 is mentioned, you are actually referring to Fig. 1b.

**Figure 2:** I would consider adding a legend to the figure to describe the red and yellow polygons. Additionally, the distance scale could be shortened as it really only needs to be up to 3km long since the image subsets are roughly 4 x 4 km.

**L214-215:** The flow of this sentence could be improved for ease of understanding. Rephrasing the sentence so that the main clause ("we applied…") starts the sentence with the rest of the information following could help. Putting some of the information into a second sentence would help as well.

**L231:** You could add some information about how the threshold was defined as four times the monthly median absolute deviation.

**L278:** This paragraph starts similarly to the last one (*place context* and *provide context*). Rephrase one of them to help it sound less repetitive.

**L278:** Make the distinction of surface melt and supraglacial lakes clearer. Describe what exactly is meant by surface melt (i.e., wetness of the entire surface vs. the pooling of meltwater into lakes).

**L278:** I believe this is the first mention of the ASCAT C-band data. Either give the description of the acronym or briefly summarize what this data is. Also include a citation/website reference for the data.

**L287-289:** Here, it is not extremely clear what the annual melt index consists of. It sounds like you are just adding up the total melt area. Is the melt intensity involved in this index? If it is just the sum of the area, how is that different than what is described in the next sentence?

**L298:** I would add "…90 winter lake drainage events" to make it clear this is not including summer drainages. Even though you say "between the 2014/2015 and 2023/2024 winter seasons", it could be interpreted that the entire years between those two seasons are included.

**L322:** One of the instances of saying "terminus of 79NG" that is incorrect. These lakes are near the grounding line, whereas the terminus is the edge of the floating tongue. The use of terminus with ZI is correct, since it lost its floating tongue, resulting in the terminus being roughly along the grounding line.

**L322-324:** I would be curious to know if the lakes fully drained both times. Or did they partially drain the first time and then finish the drainage months later? If you're able to tell that from your data, that could be interesting to include.

**L326-327:** Clarify if that means that each pixel depth across all lakes was averaged together or if the maximum or average depth of each lake was calculated and then averaged over all lakes.

**L339:** Just wanted to comment that this is a very nice display of a lot data in one figure!

**Figure 5:** Make sure the cube in km$^3$ is raised instead of using a ^ symbol.

**Figure 7:** I don't find the second subfigure (b) to be the easiest to draw information from. Consider if there is a better way to display this information. However, if you keep it as is, move the numbers to the side of the circle in cases where the circle is the same size or smaller than the number (i.e., 1, 2 and 3), to improve readability of the numbers.

**L405:** I would add "… for individual lakes over all seven years" to make the total amount of possible drainages clearer.

**L442:** Perhaps you could add more information about how the lakes are subglacially connected. Are they all along the major meltwater channel? Are some on smaller branched leading to the main channel?

**L444-459:** For Events 2-6, there is no mention on the hydrological connections of the lake clusters. Looking at the map, it does not seem that they are always connected on descending channels. Some details to that would be nice to have described here.

**L445:** Make it clear that the numbers in brackets for all following events (e.g. (~740-660m)) are referencing elevation. Upon quick glance, it could be mistaken to be the distance between them or the diameter of the lakes.

**L550:** I would make this paragraph part of the previous one.

**L551-552:** Expand on this phrase to fully address why this leads to more drainages in the winter of cool years and not in the summer, when the drainage system is presumably also inefficient.

**L572:** "…increasing the probability that meltwater reaches the grounding line, thus driving basal melting."

**L575:** Is the Zheng et al. (2023) study also in Northeast Greenland? If so, there could be a better way to combine this sentence with the previous one more fluidly.

**L594-569:** This information feels repetitive to the information at the beginning of the previous paragraph. Condense all of the information together in the previous paragraph.

**L604:** "Instead" does not feel like the right transition word here. Perhaps just saying "These near-synchronous drainages…" would help it flow better.

**L615:** "…pathway and are…"

**L640:** Would it create a better structure for this section to begin the section with this paragraph? This way, the different mechanisms are described clearly first and then can be referenced as you go through the different event descriptions. You could even label each mechanism as Mechanism 1, Mechanism 2, etc. and to make the description within each event description more concise. The way it is now, the event description paragraphs feel a little repetitive and scrambled.

**L674-684:** The flow feels a little disconnected in these two paragraphs. I would suggest reordering the sentences to something like this: First L679-682, second L674-678, third L682-684. Bringing L679-682 to the beginning gives more context to what will then be described.

**Section 5.4:** This section needs significant editing/restructuring. The first and last paragraph of this section do not provide new information; instead, they summarize information already presented in the manuscript. The first paragraph gives a nice overview of the novel method developed and implemented in this manuscript. This, however, does not belong in the discussion, but would be a nice addition at the beginning of the conclusion section. Similarly, the last paragraph addresses challenges in the presented methods. This would be more suitable to include near the end of the conclusion section. The remaining two middle paragraphs discuss the use of $\sigma°_{HV35}$ in 10-year composite imagery. The title of this section should be renamed to more specifically address the discussion of this particular part of the methodology instead of a general overview of overarching methodology of the manuscript.

**Section 5.5:** This section is also in need of restructuring in order to make the focus more precisely on future work. A majority of the sentences are used to summarize your findings or the findings and downfalls of other researchers. Much of this information has already been stated elsewhere in your manuscript and does not need to be repeated here in order to address gaps for future work. My suggestion would be to take out the few lines which describe actionable future work ideas and bring them together in a short paragraph in the conclusion section, eliminating this section entirely. In the first paragraph, this would be L721-722 about extending winter lake drainage investigations to other regions. In the second paragraph, it is the last sentence about data fusion. In the third paragraph, I do not see a clear future research statement. From the fourth paragraph, you could write a sentence about needing better or more specific data to study how winter lake drainages affect ice dynamics and how that can affect the behavior of supraglacial lakes in the melt season. In the last paragraph, you could more or less use the final sentence by exchanging "these measurements" in L751 with "SAR observations with high frequency in situ GPS measurements".

**Conclusions:** As mentioned in the last two comments, I would exchange the first paragraph here with the first paragraph from section 5.4. Then, I would put the last paragraph from section 5.4 as the second to last paragraph, leading in to the future work sentences pulled from section 5.5. The last paragraph in the conclusions section as it is can be condensed to summarize less. The sentence in L780-782 was already said at the beginning of the section and can be removed.

**Figure A1 caption:** Are the first few words in a different font?

---

## Author Comment (AC1)

**Response to reviewer 2:** We thank the reviewer for their very positive words and constructive suggestions. We agree with the suggestion regarding the velocity anomaly fields and will present the data in that way in the revised paper. We expand this point below within the Specific Comments section.

In their TCD manuscript "A decade of winter supraglacial lake drainage across Northeast Greenland using C-band SAR", Dean et al. present a database of winter supraglacial lake drainage events in Northeast Greenland derived from SAR observations. They combine data from both Sentinel-1 and the Radarsat Constellation Mission, providing a 10-year record of winter lake drainage activity. To improve temporal coverage, the authors apply a normalization approach that enables the joint use of multiple sensors and acquisition geometries. Using this merged dataset, they identify between four and eighteen drainage events per winter.

**General Comments**

Overall, the manuscript is very well written, and I enjoyed reading it. The authors successfully integrate multiple datasets and present their interpretations in a clear and well-structured way. I did not identify any fundamental flaws in the methods. However, the velocity analysis could be improved. I suggest calculating velocity anomaly fields relative to a seasonal or annual baseline velocity map. This approach would allow for a more spatially consistent analysis of velocity changes compared to interpreting arbitrarily chosen profiles.

Below, I list some specific comments that should be addressed before publication.

**Specific Comments**

**Abstract**

**L18:** Please explain what cascading events are, as not all readers may be familiar with the term.

We agree that an explanation will be useful for reviewers. We will move when we first define them (L85), to when they are first mentioned in the abstract (L8).

**L20:** You mention that there are more winter drainages when there are fewer summer drainages. Could this imply that drainage events are largely independent of the season, and instead controlled by a threshold pressure condition?

We explain the inverse relationship between lines 545-553. There are two likely reasons. In light of a comment from Reviewer 1 we plan to clarify the 2nd reason, which, as this reviewer suggests is related to the idea of threshold pressure condition.

**Introduction**

**L37:** Although you defined that "lakes" refer to supraglacial lakes earlier (L25), the phrasing reads awkwardly here. Consider using supraglacial lakes and lakes interchangeably throughout.

We agree. We will revise the definition at L26 to read "Supraglacial lakes – hereafter often referred to simply as lakes – …" to allow more flexible wording, and we will adjust the sentence around L37 for improved flow. Throughout the manuscript, we will use "supraglacial lakes" where it improves clarity/readability and "lakes" where the meaning remains unambiguous.

**L45:** WorldView imagery has a much higher resolution than 10 m, but access is limited, making it less suitable for time series analysis—perhaps better for case studies.

You raise a good point. We did not include reference to studies using high resolution sensors such as WorldView because they almost exclusively have been used for case studies rather than regional or ice sheet wide time series analysis. We will include some text briefly mentioning the use of high-resolution sensors to make the literature review more comprehensive.

**L58:** Note that Sentinel-1C is now operational.

We will rephrase this sentence to include mention of Sentinel-1C and its launch date.

**L121:** A short paragraph describing seasonal ice dynamics in the study region would improve context.

We agree that this would improve context and will include a short paragraph in the revised version.

**Methods and Data**

**Figure 1:** Please clarify what the sampling points (yellow triangles?) represent. Also, is the 10-year melt season mask derived from Landsat?

The Figure 1 caption will be revised to clarify that the yellow sampling points were used for ice velocity sampling, and that the 10-year melt season mask was derived using Landsat.

**L166:** You mention that all data were acquired in HH and HV polarization but only HV was used. Why? Schröder et al. (2020) demonstrated reduced ambiguity when combining HH and HV.

Schröder et al. (2020) indeed demonstrated that the inclusion of HH polarization can aid in the detection of supraglacial lakes. However, a primary objective of Schröder et al. (2020) was the mapping of lake area, which differs from the focus of our study. We do not attempt to delineate lake extent but instead analyze backscatter time series to detect lake drainage behaviour. We therefore chose to follow previous studies in which both HH and HV polarizations were available (e.g, Benedek & Willis, 2021; Hossain et al., 2024) and use only HV. In the context of our time-series–based approach, HH is unlikely to provide additional information, whereas HV offers superior penetration and greater sensitivity to volume scattering, making it particularly well suited for detecting winter drainage events. We do not rule out the potential usefulness of HH polarization in future methodological developments like ours, and this remains an interesting avenue for further investigation.

**Figure 2:** Does column 4 show the backscatter mean within the summer lake polygon? Please clarify. It would also help to indicate which SAR satellite (S1 or RCM) is used in panels (ii) and (iii).

Column 4 indeed shows the mean backscatter time series extracted within the summer lake polygon, and we will clarify this in the Figure 2 caption in the revised version. We note that the sensor is already indicated by the inclusion of "S1" in the titles of the panels in columns (ii) and (iii). However, for clarity, we will explicitly define the abbreviation "S1" as Sentinel-1 in the revised Figure 2 caption.

**L204:** Would using the non–terrain-corrected $\sigma^0$ values change your results?

It is likely the use of non–terrain-corrected $\sigma^0$ would result in errors associated with poorly georeferenced data. Greater spatial inconsistency of the backscatter data would result in noisier time series.

**L222:** In Figure 2 (iv), consider including side-by-side imagery from S1 and RCM for the same lake and approximate date, both before and after normalization. This could serve as a clear visual validation of your correction approach.

In the revised version we will consider including S1 and RCM imagery in the Figure, though this will be challenging as it is already quite large. We are also unsure of how useful including a comparison of both sensors would be as they were never used together in the same time series.

**L230:** See previous comment.

**L233–L234:** Please elaborate on the cause (e.g., lids?).

Thank you for pointing this out. The purpose of this additional 14-day pre/post filter is to reduce false positives by excluding transient backscatter excursions (i.e., short-lived spikes or noisy periods) and retaining only clear, sustained step-like increases in $\sigma°HV35$ (post-event minimum > pre-event maximum). Potential causes of these transient excursions include radiometric/geometric noise and residual preprocessing issues, which we described earlier (L224–226). We will add a brief clarification at L233–L234 to make this explicit.

**L237:** Please define how the end of a drainage event is determined; this is not obvious from Figure 2 (iv).

We agree that this is not clear and will be rephrased. We define the end of a drainage event as the acquisition at which the normalized HV backscatter reaches its maximum within a short transition window following the start date (up to the next three acquisitions). In the revised version, we will elaborate on our definition of how the end of the drainage event is determined. We note that the dashed vertical line shown in Figure 2 (iv) is the single representative date described at lines 240-241.

**L254:** Why was manual delineation required? Couldn't the Landsat lake masks be used here?

We found this step necessary to capture lake extent at the end of the melt season, after some lakes may have partially drained or contracted during late summer. Using Landsat-derived masks would more likely reflect near-maximum summer extents and would therefore bias our end-of-season area (and associated volume) estimates high.

**Results and Discussion**

**L365–L377:** This section is particularly interesting -- do you have a hypothesis or possible explanation for this observed behavior?

We found this to be quite interesting as well. We just present the results here in this Section 4. In our discussion (specifically Section 5.1 L540-555), we discuss two potential mechanisms that may be playing a role to explain this behavior.

**L398:** The statement seems self-evident, since summer drainages are far more numerous than winter ones.

We disagree with this point. Just because summer drainages are more numerous than winter drainages, a positive correlation between summer drainages and melt index does not necessarily mean there has to be a positive correlation between annual drainage and melt index. A negative correlation with winter drainages could offset it. The observed increase in annual drainage frequency therefore reflects a net seasonal redistribution rather than a trivial consequence of summer dominance.

**Figures 9 & 10:** These velocity plots are difficult to interpret. I suggest showing relative velocity anomalies compared to monthly or annual baselines. Importantly, note that apparent velocity increases coinciding with lake drainage (e.g., Fig. 10b ii at ~12 km) could reflect vertical displacement rather than true horizontal acceleration. SAR offset tracking cannot separate vertical and horizontal motion, and these velocity fields are not corrected for vertical effects. See for example Joughin et al. (2016) on this issue.

Thank you for the suggestion on these figures. We agree with your comment here and in general comments that showing velocity anomaly field relative to a baseline would be easier to interpret than the plots. We will include these in Figure 9 and 10 of the revised reversion. In addition, we will clarify in the revised manuscript that apparent velocity increases associated with drainage events may reflect vertical displacement rather than true horizontal acceleration, and we will review and revise related statements in the Results and Discussion accordingly. Your comment also prompted us to consider that abrupt backscatter changes associated with lake drainage may influence offset-tracking performance, and we will acknowledge this as an additional source of uncertainty when interpreting localized velocity anomalies.

**L600:** The reference to L1C is confusing; please clarify this and adjust the velocity plots accordingly.

Thank you for catching this. This is a typo. L600 should refer to L4C rather than L1C. This will be fixed in the revised version.

**L605:** If lake L5B drained first, this event would not qualify as cascading. Please clarify your terminology. You mention "basal uplift" — this may indeed be visible in the velocity fields, but again, vertical motion needs to be treated carefully (see comment above).

Given these issues, I recommend replacing profile-based analyses with velocity anomaly maps relative to an annual baseline. Such maps would better reveal spatial patterns in velocity changes and/or uplift events.

We disagree that drainage of L5B occurring first would preclude this event from being considered cascading. We use the term "cascading" to describe a chain-reaction drainage sequence, consistent with the framework of Christoffersen et al. (2018), in which drainage events may propagate upstream and/or downstream depending on stress transmission and hydrological routing. See our definition on L85 which, as we state above, we will also add to L8, the first time we refer to cascading events in the Abstract. Accordingly, if an upstream lake (L5A) subsequently drained in response to the drainage of a downstream lake (L5B), we interpret this as a coupled, chain-reaction sequence. Regarding our mention of basal uplift, this is intended as a hypothesis based on interpretation rather than a reference to the velocity fields shown in Figure 10, which, as we mention above, are horizontal velocities.

**L730–L732:** I fully agree with the statements here and the following paragraph.

We're excited to see this area of research grow.

Additional Reference

Joughin, I., Shean, D. E., Smith, B. E., & Dutrieux, P. (2016). Grounding line variability and subglacial lake drainage on Pine Island Glacier, Antarctica. Geophysical Research Letters, 43, 9093–9102. https://doi.org/10.1002/2016GL070259

---

## Author Comment (AC2)

**Response to reviewer 1:**

**General comments**

This manuscript presents a novel winter supraglacial lake drainage detection methodology using a combination of two SAR data sources. This advanced methodology normalizes for differences in acquisition geometry and satellite missions, allowing for a cohesive output with a high temporal resolution. A thorough literature review was conducted and presented in a logical manner, relating the works of other groups effectively to the topic analyzed in this manuscript. The methodology seems well thought-out and incorporates many data sources and processing chains to provide meaningful context to the drainages, in particular the volume estimation using ArcticDEM strips, subglacial water pathways mapping, and the integration of ASCAT data for surface melt estimation. The results are overall well presented and the integration of the summer drainage dataset with their winter dataset provides a novel insight into how the drainage behaviors in each season influence each other. The writing style is elevated yet easy to read; however, some restructuring and concision is needed for the final discussion sections and conclusion. Overall, I find this manuscript to be of a high quality in both methodology and presentation, with some minor work needed to clarify the details of some processes and bring the work to a succinct conclusion.

We thank the viewer for all their very positive comments. We take on board the suggestions of restructuring and concision towards the end of the paper, and of the need to clarify some of the process details. We address these general points within the specific comments below.

**Specific comments**

**L16:** "…exhibiting substantial interannual variability, *ranging from* a maximum…"

Will be fixed to include "ranging from".

**L54:** Adjust the phrase "ice layers" to more clearly reference upper ice layers

We will rephrase this so that it is clear that we are referring to upper ice layers.

**L70:** There are several instances of the British spelling of some words being used, while the majority of the manuscript is written with American English spelling. Please adjust the spellings so that they are consistent (specific examples include behaviour, analysed, favourable).

We will review the manuscript so that it is consistent with British spelling.

**L104-109:** An important fact about 79NG is that it has an extensive floating tongue. This fact should be mentioned in this paragraph and related to the study area you define in Fig. 1 (mentioning the fact that lakes that form on the floating tongue are not analyzed). The terminology used to describe your study area must also be changed to reflect the difference between the terminus and the grounding line. There are several instances throughout the manuscript where 79NG's terminus is referenced but the grounding line is meant – these two are not interchangeable for a glacier with a floating tongue and could lead to confusion by the reader.

Thank you for pointing this out. We will add information on 79NG in relation to it having a floating tongue and that lakes on it are not analyzed. We will also review the manuscript to ensure that reference to 79NG's terminus is correctly referred to as its grounding line.

**L109:** I believe where Fig. 2.1 is mentioned, you are actually referring to Fig. 1b.

You are correct. Thanks for catching this typo.

**Figure 2:** I would consider adding a legend to the figure to describe the red and yellow polygons. Additionally, the distance scale could be shortened as it really only needs to be up to 3km long since the image subsets are roughly 4 x 4 km.

We will include a legend for the red and yellow polygons and adjust the size of the scale of Figure 2.

**L214-215:** The flow of this sentence could be improved for ease of understanding. Rephrasing the sentence so that the main clause ("we applied…") starts the sentence with the rest of the information following could help. Putting some of the information into a second sentence would help as well.

Agreed. The flow will be improved in the revised version. We suggest: "We applied a multi-slope normalisation method to account for the influence of surface geophysical changes on regression-modelled incidence angle–backscatter relationships before and after drainage events."

**L231:** You could add some information about how the threshold was defined as four times the monthly median absolute deviation.

We will add some details of how it was defined.

**L278:** This paragraph starts similarly to the last one (place context and provide context). Rephrase one of them to help it sound less repetitive.

To address this, we will rephrase the opening sentences of both the paragraphs to read:

"Subglacial hydraulic potential gradients and associated flow routing were calculated to situate winter lake drainage observations within the framework of potential subglacial meltwater pathways."

And

"Daily ASCAT C-band normalised radar backscatter ($\sigma^0$) data were used to characterise the intensity and spatial extent of surface melt during each melt season across the 79NG and ZI basins."

.

**L278:** Make the distinction of surface melt and supraglacial lakes clearer. Describe what exactly is meant by surface melt (i.e., wetness of the entire surface vs. the pooling of meltwater into lakes).

We will clarify this and make the distinction in the revised version.

**L278:** I believe this is the first mention of the ASCAT C-band data. Either give the description of the acronym or briefly summarize what this data is. Also include a citation/website reference for the data.

You are correct that it is the first mention. We agree and will include a brief summary of the data and reference in the revised version. We will expand the definition as "Advanced Scatterometer (ASCAT) aboard the European Space Agency's MetOp-B satellite".

**L287-289:** Here, it is not extremely clear what the annual melt index consists of. It sounds like you are just adding up the total melt area. Is the melt intensity involved in this index? If it is just the sum of the area, how is that different than what is described in the next sentence?

Thank you for pointing this out. It is confusing. What differentiates melt intensity and index is that intensity is calculated per pixel and displayed in Figure A1, while the melt index is the sum of melt intensity of each grid cell over the study area. In our definition of the melt index, we will clarify how it is calculated and different from melt intensity.

**L298:** I would add "…90 winter lake drainage events" to make it clear this is not including summer drainages. Even though you say "between the 2014/2015 and 2023/2024 winter seasons", it could be interpreted that the entire years between those two seasons are included.

Agreed. We will include this in the revised version.

**L322:** One of the instances of saying "terminus of 79NG" that is incorrect. These lakes are near the grounding line, whereas the terminus is the edge of the floating tongue. The use of terminus with ZI is correct, since it lost its floating tongue, resulting in the terminus being roughly along the grounding line.

Thank you for pointing this out. We will correct it to specify that it was near the grounding line of 79NG rather than the terminus.

**L322-324:** I would be curious to know if the lakes fully drained both times. Or did they partially drain the first time and then finish the drainage months later? If you're able to tell that from your data, that could be interesting to include.

We were also interested in this behaviour. While we cannot determine this definitively, we suspect that these lakes partially drained and subsequently completed drainage several months later. This interpretation is based on the backscatter time series, which shows an initial increase followed by a prolonged stable period with little to no change, and then a second increase. If the initial increase had been followed by a gradual decrease in backscatter, this might suggest winter refilling mechanism. However, we do not observe this in the backscatter time series and therefore consider winter lake filling to be unlikely.

**L326-327:** Clarify if that means that each pixel depth across all lakes was averaged together or if the maximum or average depth of each lake was calculated and then averaged over all lakes.

We think this is clear as we make no mention of maximum depth and refer only to mean depths. Whether we take the mean of all pixel depths across all lakes, or whether we take the mean depths of each lake and then take the mean of those, the result will be the same.

**L339:** Just wanted to comment that this is a very nice display of a lot data in one figure!

Thank you!

**Figure 5:** Make sure the cube in km3 is raised instead of using a ^ symbol.

We will fix this.

**Figure 7:** I don't find the second subfigure (b) to be the easiest to draw information from. Consider if there is a better way to display this information. However, if you keep it as is, move the numbers to the side of the circle in cases where the circle is the same size or smaller than the number (i.e., 1, 2 and 3), to improve readability of the numbers.

For the revised version, we will consider other options to display the data. If we decide to keep it as is, numbers will be moved to the side as suggested.

**L405:** I would add "… for individual lakes *over all seven years*" to make the total amount of possible drainages clearer.

We will include this.

**L442:** Perhaps you could add more information about how the lakes are subglacially connected. Are they all along the major meltwater channel? Are some on smaller branched leading to the main channel?

We agree it would be beneficial to include this detail and will include it in the revised version. Information on whether lakes are located along tributaries or the main channel will be described.

**L444-459:** For Events 2-6, there is no mention on the hydrological connections of the lake clusters. Looking at the map, it does not seem that they are always connected on descending channels. Some details to that would be nice to have described here.

We will include mention of the hydrologic connection for these events as was done for the first event.

**L445:** Make it clear that the numbers in brackets for all following events (e.g. (~740-660m)) are referencing elevation. Upon quick glance, it could be mistaken to be the distance between them or the diameter of the lakes.

We will clarify this.

**L550:** I would make this paragraph part of the previous one.

We agree and will change this.

**L551-552:** Expand on this phrase to fully address why this leads to more drainages in the winter of cool years and not in the summer, when the drainage system is presumably also inefficient.

We will expand this point in the revised version to fully address this.

**L572:** "…increasing the probability that meltwater reaches the *grounding line, thus driving basal melting.*"

We will change our text to this.

**L575:** Is the Zheng et al. (2023) study also in Northeast Greenland? If so, there could be a better way to combine this sentence with the previous one more fluidly.

Their study was ice sheet wide; however, the value reported was specifically for Northeast Greenland. We will consider combining the sentences in the revised version.

**L594-569:** This information feels repetitive to the information at the beginning of the previous paragraph. Condense all of the information together in the previous paragraph.

We will condense this information into the previous paragraph.

**L604:** "Instead" does not feel like the right transition word here. Perhaps just saying "These near synchronous drainages…" would help it flow better.

We will incorporate this suggestion to improve flow.

**L615:** "…pathway *and* are…"

This fix will be included.

**L640:** Would it create a better structure for this section to begin the section with this paragraph? This way, the different mechanisms are described clearly first and then can be referenced as you go through the different event descriptions. You could even label each mechanism as Mechanism 1, Mechanism 2, etc. and to make the description within each event description more concise. The way it is now, the event description paragraphs feel a little repetitive and scrambled.

We agree that moving this paragraph to the beginning of Section 5.2 will improve the overall structure. We will consider labelling the mechanisms to reduce repetitiveness throughout the paragraphs.

**L674-684:** The flow feels a little disconnected in these two paragraphs. I would suggest reordering the sentences to something like this: First L679-682, second L674-678, third L682-684. Bringing L679-682 to the beginning gives more context to what will then be described.

We agree and will restructure this accordingly in the revised version.

**Section 5.4:** This section needs significant editing/restructuring. The first and last paragraph of this section do not provide new information; instead, they summarize information already presented in the manuscript. The first paragraph gives a nice overview of the novel method developed and implemented in this manuscript. This, however, does not belong in the discussion, but would be a nice addition at the beginning of the conclusion section. Similarly, the last paragraph addresses challenges in the presented methods. This would be more suitable to include near the end of the conclusion section. The remaining two middle paragraphs discuss the use of $\sigma°HV35$ in 10-year composite imagery. The title of this section should be renamed to more specifically address the discussion of this particular part of the methodology instead of a general overview of overarching methodology of the manuscript.

**Section 5.5:** This section is also in need of restructuring in order to make the focus more precisely on future work. A majority of the sentences are used to summarize your findings or the findings and downfalls of other researchers. Much of this information has already been stated elsewhere in your manuscript and does not need to be repeated here in order to address gaps for future work. My suggestion would be to take out the few lines which describe actionable future work ideas and bring them together in a short paragraph in the conclusion section, eliminating this section entirely. In the first paragraph, this would be L721-722 about extending winter lake drainage investigations to other regions. In the second paragraph, it is the last sentence about data fusion. In the third paragraph, I do not see a clear future research statement. From the fourth paragraph, you could write a sentence about needing better or more specific data to study how winter lake drainages affect ice dynamics and how that can affect the behavior

of supraglacial lakes in the melt season. In the last paragraph, you could more or less use the final sentence by exchanging "these measurements" in L751 with "SAR observations with high frequency in situ GPS measurements".

**Conclusions:** As mentioned in the last two comments, I would exchange the first paragraph here with the first paragraph from section 5.4. Then, I would put the last paragraph from section 5.4 as the second to last paragraph, leading in to the future work sentences pulled from section 5.5. The last paragraph in the conclusions section as it is can be condensed to summarize less. The sentence in L780-782 was already said at the beginning of the section and can be removed. Figure A1 caption: Are the first few words in a different font?

We agree that Sections 5.4 and 5.5 contain some repetitive material and would benefit from restructuring. For Section 5.4, we will move the content of the first and last paragraphs to the Conclusions section, as suggested, and revise the section title to more specifically reflect the discussion of the $\sigma°HV$-based composite methodology. We will eliminate Section 5.5 and move lines on actionable future research directions into the conclusion.

Regarding the Figure A1 caption, there indeed appears to be a different font. We will fix this.